# UNIFIED-IO: A UNIFIED MODEL FOR VISION, LANGUAGE, AND MULTI-MODAL TASKS

**Jiasen Lu**[†,*] **Christopher Clark**[†*], **Rowan Zellers**[†◇],

**Roozbeh Mottaghi**[†◇], **Aniruddha Kembhavi**[†◇]

[†]Allen Institute for AI, [◇]University of Washington, Seattle

## ABSTRACT

We propose UNIFIED-IO, a model that performs a large variety of AI tasks spanning classical computer vision tasks, including pose estimation, object detection, depth estimation and image generation, vision-and-language tasks such as region captioning and referring expression, to natural language processing tasks such as question answering and paraphrasing. Developing a single unified model for such a large variety of tasks poses unique challenges due to the heterogeneous inputs and outputs pertaining to each task, including RGB images, per-pixel maps, binary masks, bounding boxes, and language. We achieve this unification by homogenizing every supported input and output into a sequence of discrete vocabulary tokens. This common representation across all tasks allows us to train a single transformer-based architecture, jointly on over 90 diverse datasets in the vision and language fields. UNIFIED-IO is the first model capable of performing all 7 tasks on the GRIT benchmark and produces strong results across 16 diverse benchmarks like NYUv2-Depth, ImageNet, VQA2.0, OK-VQA, Swig, VizWiz-Ground, BoolQ, and SciTail, with no task-specific fine-tuning. Code and demos for UNIFIED-IO are available at: `unified-io.allenai.org`

## 1 INTRODUCTION

We present UNIFIED-IO, the first neural model to jointly perform a large and diverse set of AI tasks spanning classical computer vision (such as object detection, segmentation, and depth estimation), image synthesis (such as image generation and image in-painting), vision-and-language (like visual question answering, image captioning, and referring expression) and NLP (such as question answering and paraphrasing). Unified general-purpose models avoid the need for task-specific design, learn and perform a wide range of tasks with a single architecture, can utilize large, diverse data corpora, can effectively transfer concept knowledge across tasks, and even perform tasks unknown and unobserved at design and training time.

Building unified models for computer vision has proven to be quite challenging since vision tasks have incredibly diverse input and output representations. For instance, object detection produces bounding boxes around objects in an image, segmentation produces binary masks outlining regions in an image, visual question answering produces an answer as text, and depth estimation produces a map detailing the distance of each pixel from the camera. This heterogeneity makes it very challenging to architect a single model for all these tasks. In contrast, while the landscape of natural language processing (NLP) tasks, datasets, and benchmarks is large and diverse, their inputs and desired outputs can often be uniformly represented as sequences of tokens. Sequence to sequence (Seq2Seq) architectures (Raffel et al., 2020; Brown et al., 2020), specifically designed to accept and produce such sequences of tokens, are thus widely applicable to many tasks. Unified models employing such architectures have been central to much recent progress in NLP.

Unified models for computer vision typically use a shared visual backbone to produce visual embeddings but then employ individual branches for each of the desired tasks. These include models

---

[*]Equal contribution. Correspondence to `jiasenl@allenai.org`

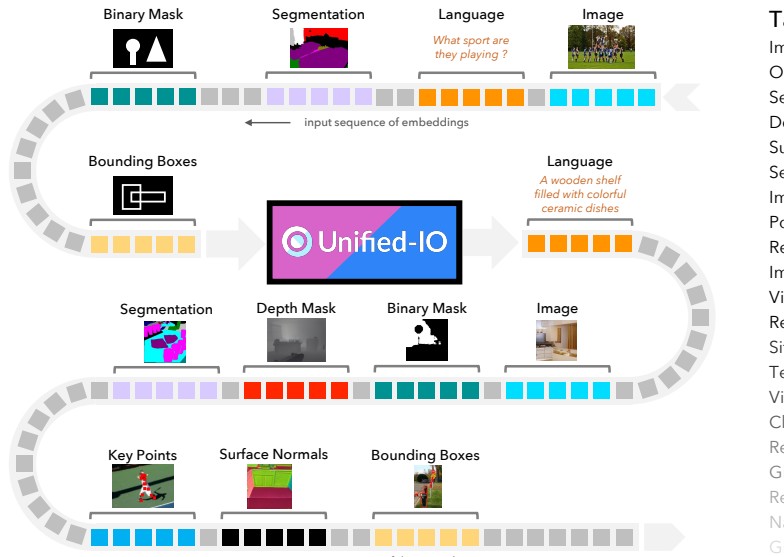

Figure 1: UNIFIED-IO is a single sequence-to-sequence model that performs a variety of tasks in computer vision and NLP using a unified architecture without a need for either task or modality-specific branches. This broad unification is achieved by homogenizing every task's input and output into a sequence of discrete vocabulary tokens. UNIFIED-IO supports modalities as diverse as images, masks, keypoints, boxes, and text, and tasks as varied as depth estimation, inpainting, semantic segmentation, captioning, and reading comprehension.

like Mask R-CNN (He et al., 2017) for classical visual tasks that use an ImageNet pre-trained encoder followed by branches for detection and segmentation, trained in a fully supervised manner. In the vision and language (V&L) domain, CNN backbones feed visual features to transformer architectures that also combine language, followed by task-specific heads for visual question answering, referring expression, visual commonsense reasoning, etc. (Lu et al., 2019; Li et al., 2019; Tan & Bansal, 2019). A more recent trend has seen the emergence of unified architectures that do away with task-specific heads and instead introduce modality-specific heads (Hu & Singh, 2021; Cho et al., 2021; Gupta et al., 2022a; Wang et al., 2022b) – for instance, a single language decoder that serves multiple tasks requiring language output like captioning and classification. However, most progress in unified models continues to be centered around V&L tasks, owing to the simplicity of building shared language decoders and is often limited to supporting just a handful of tasks.

UNIFIED-IO is a Seq2Seq model capable of performing a variety of tasks using a unified architecture without a need for either task or even modality-specific branches. This broad unification is achieved by homogenizing every task's output into a sequence of discrete tokens. Dense structured outputs such as images, segmentation masks and depth maps are converted to sequences using a vector quantization variational auto-encoder (VQ-GAN) (Esser et al., 2021), sparse structured outputs such as bounding boxes, and human joint locations are transcribed into sequences of coordinate tokens, and language outputs are converted to sequences using byte-pair encoding. This unification enables Unified-IO to jointly train on over 90 datasets spanning computer vision, V&L, and NLP tasks with a single streamlined transformer encoder-decoder architecture (Raffel et al., 2020).

Our jointly trained UNIFIED-IO is the first model to support all 7 tasks in the General Robust Image Task (GRIT) Benchmark (Gupta et al., 2022b) and obtains the top overall score of 64.3 when averaging across all tasks, handily beating the second best model by 32.0. We further evaluate UNIFIED-IO on 16 diverse benchmarks across computer vision and NLP, without any fine-tuning towards any individual benchmark, and find that it performs remarkably well compared to specialized (or fine-tuned) state-of-the-art models.

## 2  VISION, LANGUAGE AND MULTI-MODAL TASKS

UNIFIED-IO is designed to handle a wide range of language, vision and language, and classic vision tasks in a unified way. To fully test this capability, we gather 95 vision, language, and multi-modal

| | Example Source | Size | | | | Input Modalities | | | | Output Modalities | | | |
|---|---|---|---|---|---|---|---|---|---|---|---|---|---|
| | | Datasets | Size | Percent | Rate | Text | Image | Sparse | Dense | Text | Image | Sparse | Dense |
| **Image Synthesis** | | **14** | **56m** | **43.0** | **18.7** | ✓ | ✓ | ✓ | ✓ | - | ✓ | - | - |
| Image Synthesis from Text | *RedCaps* | 9 | 55m | 41.9 | 16.7 | ✓ | - | - | - | - | ✓ | - | - |
| Image Inpainting | *VG* | 3 | 1.2m | 0.9 | 1.5 | ✓ | ✓ | ✓ | - | - | ✓ | - | - |
| Image Synthesis from Seg. | *LVIS* | 2 | 220k | 0.2 | 0.6 | ✓ | - | - | ✓ | - | ✓ | - | - |
| **Sparse Labelling** | | **10** | **8.2m** | **6.3** | **12.5** | ✓ | ✓ | ✓ | - | - | - | ✓ | - |
| Object Detection | *Open Images* | 3 | 1.9m | 1.5 | 3.6 | - | ✓ | - | - | - | - | ✓ | - |
| Object Localization | *VG* | 3 | 6m | 4.6 | 7.1 | ✓ | ✓ | - | - | - | - | ✓ | - |
| Keypoint Estimation | *COCO* | 1 | 140k | 0.1 | 0.7 | - | ✓ | ✓ | - | - | - | ✓ | - |
| Referring Expression | *RefCoco* | 3 | 130k | 0.1 | 1.1 | ✓ | ✓ | - | - | - | - | ✓ | - |
| **Dense Labelling** | | **6** | **2.4m** | **1.8** | **6.2** | ✓ | ✓ | - | - | - | - | - | ✓ |
| Depth Estimation | *NYU Depth* | 1 | 48k | 0.1 | 0.4 | - | ✓ | - | - | - | - | - | ✓ |
| Surface Normal Estimation | *Framenet* | 2 | 210k | 0.2 | 1.1 | - | ✓ | - | - | - | - | - | ✓ |
| Object Segmentation | *LVIS* | 3 | 2.1m | 1.6 | 4.7 | ✓ | ✓ | - | - | - | - | - | ✓ |
| **Image Classification** | | **9** | **22m** | **16.8** | **12.5** | - | ✓ | ✓ | - | ✓ | - | - | - |
| Image Classification | *ImageNet* | 6 | 16m | 12.2 | 8.1 | ✓ | ✓ | - | - | ✓ | - | - | - |
| Object Categorization | *COCO* | 3 | 6m | 4.6 | 4.4 | - | ✓ | ✓ | - | ✓ | - | - | - |
| **Image Captioning** | | **7** | **31m** | **23.7** | **12.5** | - | ✓ | ✓ | - | ✓ | - | - | - |
| Webly Supervised Captioning | *CC12M* | 3 | 26m | 19.7 | 8.8 | - | ✓ | - | - | ✓ | - | - | - |
| Supervised Captioning | *VizWiz* | 3 | 1.4m | 1.1 | 1.7 | - | ✓ | - | - | ✓ | - | - | - |
| Region Captioning | *VG* | 1 | 3.8m | 2.9 | 2.0 | - | ✓ | ✓ | - | ✓ | - | - | - |
| **Vision & Language** | | **16** | **4m** | **3.0** | **12.5** | ✓ | ✓ | ✓ | - | ✓ | - | - | ✓ |
| Visual Question Answering | *VQA 2.0* | 13 | 3.3m | 2.5 | 10.4 | ✓ | ✓ | ✓ | - | ✓ | - | - | - |
| Relationship Detection | *VG* | 2 | 640k | 0.5 | 1.9 | - | ✓ | ✓ | - | ✓ | - | - | - |
| Grounded VQA | *VizWiz* | 1 | 6.5k | 0.1 | 0.1 | ✓ | ✓ | - | - | ✓ | - | - | ✓ |
| **NLP** | | **31** | **7.1m** | **5.4** | **12.5** | ✓ | - | - | - | ✓ | - | - | - |
| Text Classification | *MNLI* | 17 | 1.6m | 1.2 | 4.8 | ✓ | - | - | - | ✓ | - | - | - |
| Question Answering | *SQuAD* | 13 | 1.7m | 1.3 | 5.2 | ✓ | - | - | - | ✓ | - | - | - |
| Text Summarization | *Gigaword* | 1 | 3.8m | 2.9 | 2.5 | ✓ | - | - | - | ✓ | - | - | - |
| **Language Modelling** | | **2** | **-** | **-** | **12.5** | ✓ | - | - | - | ✓ | - | - | - |
| Masked Language Modelling | *C4* | 2 | - | - | 12.5 | ✓ | - | - | - | ✓ | - | - | - |
| **All Tasks** | | **95** | **130m** | **100** | **100** | ✓ | ✓ | ✓ | ✓ | ✓ | ✓ | ✓ | ✓ |

Table 1: Tasks UNIFIED-IO learns to complete. From left to right, columns show an example of one of the sources used for the task, the number of datasets, total number and percent of examples relative to the entire training corpus, and sample rate during multi-task training. Subsequent columns show what modalities are required for the tasks, and highlighted rows show aggregated statistics for groups of similar tasks.

datasets from 62 publicly available data sources as targets for our model to learn during multi-task training. These datasets cover a wide range of tasks, skills, and modalities.

We categorize the input and output modalities of each task into 4 different types: Text – natural language tokens; Image – RGB images; Sparse – a small number of location coordinates within the image; Dense – per-pixel labels such as depth maps, surface normal maps, *etc*. We group related datasets into 8 groups and 22 tasks to facilitate our training and analysis:

**Image Synthesis.** Given a text description, partially occluded image and inpainting target, or segmentation map containing a semantic class for some pixels, generate a matching image. Data sources with image and text pairs (Desai et al., 2021), bounding boxes (Krishna et al., 2017) or semantic segmentation (Gupta et al., 2019) can be used to build these tasks.

**Sparse Labelling.** Given an image and a natural language query, identify the target regions or keypoint locations that are being referred to. Tasks include object detection (Kuznetsova et al., 2020), object localization (Rhodes et al., 2017), human pose estimation (Lin et al., 2014), and referring expression (Kazemzadeh et al., 2014).

**Dense Labelling.** Given an image, produce per-pixel labels for that image. Labels include the distance of that pixel to the camera (Nathan Silberman & Fergus, 2012), surface orientation (Bae et al., 2021) or semantic class (Lin et al., 2014).

**Image Classification.** Given an image and optionally a target bounding box, generate a class name or tag of that image or target region. This group includes image classification (Deng et al., 2009) and object categorization (Pinz et al., 2006) datasets.

**Image Captioning.** Given an image and optionally a bounding box, generate a natural language description of that image or target region. We include both crowd-sourced (Chen et al., 2015) and webly supervised (Changpinyo et al., 2021) captions.

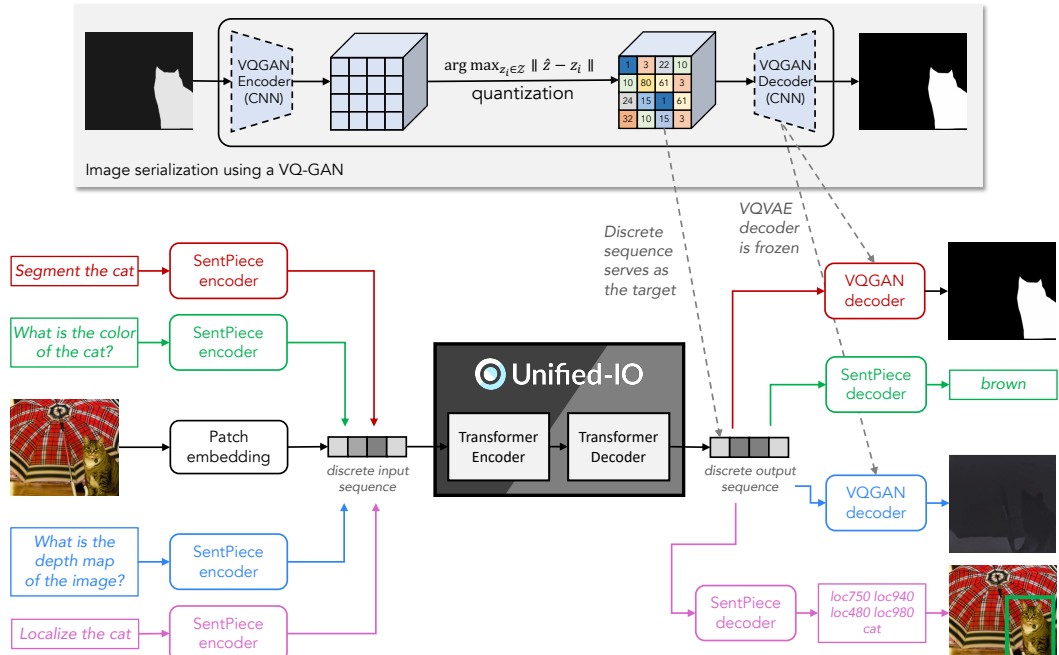

Figure 2: **Unified-IO.** A schematic of the model with four demonstrative tasks: object segmentation, visual question answering, depth estimation and object localization.

**Vision & Language.** A broad category for other tasks that require joint reason over image content and a natural language query. There are many popular vision and language datasets, and we categories these datasets into 3 tasks – visual question answering (Antol et al., 2015); relationship detection (Lu et al., 2016) and grounded VQA (Chen et al., 2022a).

**NLP.** Tasks with text as the only input and output modalities, including text classification (Williams et al., 2018), question answering (Rajpurkar et al., 2016) and text summarization (Graff et al., 2003).

**Language Modeling.** The masking language modeling pre-training task (See Section 3.3) using text from C4 (Raffel et al., 2020) and Wikipedia (Foundation), which we include to ensure the knowledge gained from language pre-training is not lost during multi-task training. Other pre-training tasks are not included because the relevant datasets are already used in other supervised tasks (*e.g.*, for captioning or classification).

Table 1 shows the details of tasks and groups. We list an example dataset source, number of datasets, number of examples, percent of the total number of examples, and sampling rate during training (Section 3.3) for each group and task. Subsequent columns show what modalities are required for the inputs and outputs. We defer additional task details, inference details, the complete list of datasets and visualizations to the Appendix A.1.

## 3 UNIFIED-IO

Our goal is to build a single unified model that can support a diverse set of tasks across computer vision and language with little to no need for task-specific customizations and parameters. Such unified architectures can be applied to new tasks with little to no knowledge of the underlying machinery, enable general pre-training to benefit many diverse downstream applications, be jointly trained on a large number of tasks, and better allows knowledge to be shared between tasks.

### 3.1 UNIFIED TASK REPRESENTATIONS

Supporting a variety of modalities such as images, language, boxes, binary masks, segmentation masks, *etc.* without task-specific heads requires representing these modalities in a shared and unified space. To do this, we discretize the text, images, and other structured outputs in our tasks and represent them with tokens drawn from a unified and finite vocabulary.

**Text representation.** Following Raffel et al. (2020), text inputs and outputs are tokenized using SentencePiece (Kudo & Richardson, 2018). Following past works such as McCann et al. (2018); Raffel et al. (2020); Gupta et al. (2022a); Wang et al. (2022b) we also specify each task with a natural language prompt (excluding some tasks like VQA, which are fully specified by their text inputs) in order to indicate what task should be performed. For example, "*What is the depth map of the image?*" for depth estimation or "*What region does "cat" describe?*" for object localization.

**Images and dense structures representation.** A variety of tasks in computer vision requires the model to produce high-dimensional outputs such as images (*e.g.*, image in-painting) or per-pixel labels (*e.g.*, depth estimation). To handle these modalities, we first convert per-pixel labels into RGB images. For depth, we construct a grayscale image by normalizing the depth map. For surface normal estimation, we convert the $x/y/z$ orientations into $r/g/b$ values. For segmentation, we map each instance present in the image to a unique color. We randomly select colors for each instance and specify the color-to-class mapping in the text instead of using universal color-to-class mapping. This avoids requiring a fixed list of classes and avoids having colors that may only be marginally different due to the presence of a large number of classes.

Then we encode these images as discrete tokens using a VQ-GAN. In particular, we use the imagenet-pretrained VQ-GAN from Esser et al. (2021) with $256 \times 256$ resolution, a compression ratio of 16, and 16384 codebook size. The VQ-GAN codebook is added to the vocabulary as additional tokens that can be generated by the decoder. During training, the tokens for the target image are used as targets. During inference, the VQ-GAN decoder is used to convert the generated image tokens into an output image.

**Sparse structures representation.** We encode sparse structures such as bounding boxes or human joints by adding 1000 special tokens to the vocabulary to represent discretized image coordinates (Chen et al., 2022b). Points are then encoded with a sequence of two such tokens, one for the $x$ and one for the $y$ coordinates, and boxes are encoded using a sequence of four tokens, two for the upper right corner and two for the lower left corner. Labeled boxes are encoded as a box followed by a text class label, and joints are encoded as a sequence of points followed by a text visibility label. This allows us to handle a wide variety of tasks that use these elements in their inputs or output (see Appendix A.1 for examples).

## 3.2 UNIFIED ARCHITECTURE

Universally representing a wide variety of tasks as input and output sequences of discrete tokens enables us to employ architectures that have been proven successful in natural language processing. In UNIFIED-IO, we propose a pure transformer model largely following the design of T5 (Raffel et al., 2020). In particular, UNIFIED-IO is an encoder-decoder architecture where both the encoder and decoder are composed of stacked transformer layers, which in turn are composed of self-attention transformers, cross-attention transformers (in the decoder), and feed-forward neural networks. The layers are applied residually, and layer norms are applied before each transformer and feed-forward network. See Raffel et al. (2020) for details.

We make a few architectural changes to adapt the T5 architecture to our setting. First, to handle input images, we reshape the image into a sequence of patches that are embedded with linear projection similar to Dosovitskiy et al. (2021). Second, we expand the vocabulary to include the location tokens and the image tokens used in the VQ-GAN. Third, we extend the 1-d relative embedding (Dosovitskiy et al., 2021) to 2-d with a fixed number of learned embeddings. We also add absolute position embedding to the token embedding following Devlin et al. (2019), since the absolute position information is essential to image tasks.

We use a maximum of 256 and 128 text tokens for inputs and outputs respectively, and a maximum length of 576 (*i.e.* $24 \times 24$ patch encoding from a $384 \times 384$ image) for image inputs and 256 (*i.e.* $16 \times 16$ latent codes from a $256 \times 256$ image) for image outputs. In this work, we present four versions of UNIFIED-IO ranging from 71 million to 2.9 billion parameters, as detailed in Table 2.

## 3.3 TRAINING

UNIFIED-IO is trained in two stages – A pre-training stage that uses unsupervised losses from text, image, and paired image-text data, and a massive multi-task stage where the model is jointly trained

| Model | Encoder Layers | Decoder Layers | Model Dims | MLP Dims | Heads | Total Params |
|---|---|---|---|---|---|---|
| UNIFIED-IO$_{\text{SMALL}}$ | 8 | 8 | 512 | 1024 | 6 | 71M |
| UNIFIED-IO$_{\text{BASE}}$ | 12 | 12 | 768 | 2048 | 12 | 241M |
| UNIFIED-IO$_{\text{LARGE}}$ | 24 | 24 | 1024 | 2816 | 16 | 776M |
| UNIFIED-IO$_{\text{XL}}$ | 24 | 24 | 2048 | 5120 | 32 | 2925M |

Table 2: Size variant of UNIFIED-IO. Both encoder and decoder are based on T5 implementation (Raffel et al., 2020). Parameters of VQ-GAN (Esser et al., 2021) are not included in the total parameter count.

on a large variety of tasks. Since our goal is to examine whether a single unified model can solve a variety of tasks simultaneously, we **do not perform task-specific fine-tuning** although prior work (Lu et al., 2020; Wang et al., 2022b) shows it can further improve task performance.

**Pre-training.** To learn good representations from large-scale webly supervised image and text data, we consider two pre-training tasks: *text span denoising* and *masked image denoising*. The text span denoising task follows Raffel et al. (2020) – randomly corrupt 15% of the tokens and replace the consecutive corrupted tokens with a unique mask token. The masked image denoising task follows Bao et al. (2022) and He et al. (2022) – randomly masked 75% of the image patches, and the goal is to recover the whole image. When another modality is present, *i.e.* image or text, the model can use information from that modality to complete the tasks.

We construct the pre-training dataset by incorporating publicly available language data (i.e., plain texts from Common Crawl), vision data (i.e., raw images from different datasets), and V&L data (i.e., image caption and image label pairs). For V&L data, we add a simple prompt "*An image of*" at the beginning of the caption or categories to indicate it is multi-modal data (Wang et al., 2022d).

We pre-train UNIFIED-IO on this combination of datasets with an in-batch mixing strategy. We equally sample data with the text and image denoising objective. For text denoising, half of the samples are from pure text data, *i.e.* C4 and Wikipedia. The other half is constructed from image and class data, such as Imagenet21k (Ridnik et al., 2021) or image and caption data, such as YFCC15M (Radford et al., 2021). For image denoising, we also use the same caption and class data and some image-only data from datasets for our vision tasks. We sample from datasets in proportion to dataset size. See Appendix A.2 for details.

**Multi-tasking.** To build a single unified model for diverse vision, language, and V&L tasks, we construct a massive multi-tasking dataset by ensembling 95 datasets from 62 publicly available data sources. See Section 2 for task details and Appendix A.1 for dataset visualizations.

We jointly train UNIFIED-IO on this large set of datasets by mixing examples from these datasets within each batch. We equally sample each group (1/8) except for image synthesis (3/16) and dense labeling (1/16) since dense labeling has significantly fewer data and image synthesis has significantly more data than other groups. Within each group, we sample datasets proportional to the square root of their size to better expose the model to underrepresented tasks. Due to the large variance in dataset size, some tasks are still rarely sampled (*e.g.* depth estimation only has a $0.43\%$ chance of being sampled). See Appendix A.3 for details and visualizations.

**Implementation Details.** Due to space limitation, see Appendix A.4 for implementation details.

## 4 EXPERIMENTS

We now present results for UNIFIED-IO on the GRIT benchmark (Sec 4.1), ablate training data via the GRIT ablation benchmark (Sec 4.2) and evaluate UNIFIED-IO on 16 other benchmarks in computer vision and NLP (Sec 4.3). Appendix A.5 shows the evaluation of the same concept and new concept on GRIT and A.6 shows the prompt generalization. Qualitative examples are in A.9.

### 4.1 RESULTS ON GRIT

The General Robust Image Task (GRIT) Benchmark (Gupta et al., 2022b) is an evaluation-only benchmark designed to measure the performance of models across multiple tasks, concepts, and data sources. GRIT aims to encourage the building of unified and general purpose vision models and is thus well suited to evaluate UNIFIED-IO. GRIT has seven tasks that cover a range of visual skills with varying input and output modalities and formats: categorization, localization, VQA, refer expression, segmentation, keypoint, and surface normal estimation.

| | Categorization | | Localization | | VQA | | Refexp | | Segmentation | | Keypoint | | Normal | | All | |
|---|---|---|---|---|---|---|---|---|---|---|---|---|---|---|---|---|
| | ablation | test | ablation | test | ablation | test | ablation | test | ablation | test | ablation | test | ablation | test | ablation | test |
| 0 NLL-AngMF [4] | - | - | - | - | - | - | - | - | - | - | - | - | **49.6** | **50.5** | 7.2 | 7.1 |
| 1 Mask R-CNN [42] | - | - | 44.7 | 45.1 | - | - | - | - | 26.2 | 26.2 | **70.8** | **70.6** | - | - | 20.2 | 20.3 |
| 2 GPV-1 [39] | 33.2 | 33.2 | 42.8 | 42.7 | 50.6 | 49.8 | 25.8 | 26.8 | - | - | - | - | - | - | 21.8 | 21.8 |
| 3 CLIP [87] | 48.1 | - | - | - | - | - | - | - | - | - | - | - | - | - | 6.9 | - |
| 4 OFA_LARGE [108] | 22.6 | - | - | - | 72.4 | - | 61.7 | - | - | - | - | - | - | - | 22.4 | - |
| 5 GPV-2 [53] | 54.7 | 55.1 | 53.6 | 53.6 | 63.5 | 63.2 | 51.5 | 52.1 | - | - | - | - | - | - | 31.9 | 32.0 |
| 6 UNIFIED-IO_SMALL | 42.6 | - | 50.4 | - | 52.9 | - | 51.1 | - | 40.7 | - | 46.5 | - | 33.5 | - | 45.4 | - |
| 7 UNIFIED-IO_BASE | 53.1 | - | 59.7 | - | 63.0 | - | 68.3 | - | 49.3 | - | 60.2 | - | 37.5 | - | 55.9 | - |
| 8 UNIFIED-IO_LARGE | 57.0 | - | 64.2 | - | 67.4 | - | 74.1 | - | 54.0 | - | 67.6 | - | 40.2 | - | 60.7 | - |
| 9 UNIFIED-IO_XL | **61.7** | **60.8** | **67.0** | **67.1** | **74.5** | **74.5** | **78.6** | **78.9** | **56.3** | **56.5** | 68.1 | 67.7 | 45.0 | 44.3 | **64.5** | **64.3** |

Table 3: Comparison of our UNIFIED-IO models to recent SOTA on GRIT benchmark. UNIFIED-IO is the first model to support all seven tasks in GRIT. Results of CLIP, OFA obtained from GRIT challenge.

UNIFIED-IO is the first model to support all seven tasks in GRIT. As seen in Table 3, UNIFIED-IO_XL outperforms all prior submissions to GRIT obtaining average accuracy of 64.3 on test. The next best submission is GPV-2 (Kamath et al., 2022) which obtains 32.0 and can only support 4 out of 7 tasks. UNIFIED-IO_XL also outperforms the multi-task checkpoint of OFA_LARGE (Wang et al., 2022b) on VQA, refer expression and categorization.

Mask R-CNN (He et al., 2017) is a strong baseline for core vision tasks. UNIFIED-IO_XL outperforms Mask R-CNN on localization and segmentation. The reason is UNIFIED-IO_XL shows little degradation in performance between the same concept and the new concept as discussed in Appendix A.5. On keypoint, our model is worse compared to Mask R-CNN (68.1 *vs.* 70.8). The reason is we have 2-stage inference for keypoint – first locate the person using the object localization prompt, then find keypoints for each detected region.

NLL-AngMF (Bae et al., 2021) is a SOTA model for surface normal estimation. Our model gets strong results compared to NLL-AngMF (44.3 *vs.* 49.6). Since our image tokenizer is only pre-trained on ImageNet without any surface normal data, the upper bound of our method through reconstruction is 59.8 on FrameNet (Kazemzadeh et al., 2014). This suggests our score could be considerably improved by training a stronger image tokenizer.

## 4.2 ABLATIONS

To better understand how multi-tasking affects learning, we perform ablations by leaving out individual task groups from multi-task training. Due to computational constraints, we ablate UNIFIED-IO_LARGE and train for 250k steps. If ablating a task group, we reduce the number of training steps so that all models are trained on approximately the same number of examples for each of the remaining task groups. Results are shown in Table 4 on GRIT and MNLI (Williams et al., 2018).

In spite of supporting a large number of heterogeneous tasks, Unified-IO is able to perform well across all tasks. Reducing this heterogeneity by removing task groups does not impact the performance of individual tasks significantly. This is notable since removing a task group significantly reduces the scope of what a model needs to learn while keeping the model capacity fixed. This empirically demonstrates the effectiveness of the proposed unified architecture for massive heterogeneous task support.

An exception is that removing the NLP group significantly boosts categorization, which might indicate that the sentence classification task interferes with image classification. Removing captioning also boosts performances on VQA and a few other tasks, which might be caused by captioning requiring a relatively large amount of model capacity to learn free-form text generation, in contrast to VQA that requires short answer phrases from a limited vocabulary. Removing image synthesis causes a major regression in keypoint. Manual inspection shows that the model predicts standing-human shaped keypoints even for people in very different postures, suggesting the model learned to rely on priors instead of the image content. We also see minor regressions in localization and referring expression, suggesting that image synthesis tasks, possibly image in-painting in particular, had a surprising positive transfer to understanding sparse structured outputs. It is possible that an ablation analysis on the XL model may yield different outcomes, but we are unable to perform an XL-based analysis due to limited compute.

| Model | Step | Categorization | Localization | VQA | Refexp | Segmentation | Keypoint | Normal | MNLI |
|---|---|---|---|---|---|---|---|---|---|
| UNIFIED-IO_LARGE | 250k | 50.3 | 63.4 | 65.7 | 73.4 | 51.8 | 69.2 | 40.7 | 85.1 |
| w/o Image Synthesis | 200k | 52.7 | 62.9 | 64.2 | 72.0 | 53.6 | 18.3 | **42.2** | 84.3 |
| w/o Sparse | 220k | 52.6 | - | 64.1 | - | 51.3 | - | 38.5 | 83.8 |
| w/o Dense | 235k | 49.5 | 62.4 | 65.6 | 72.9 | - | 66.7 | - | 84.8 |
| w/o Classification | 220k | - | 63.1 | 64.0 | 73.7 | 52.1 | 66.8 | 39.1 | 84.6 |
| w/o Captioning | 220k | 49.7 | **65.0** | **68.0** | **74.7** | **54.2** | 67.4 | 39.2 | **85.3** |
| w/o V&L | 220k | 50.9 | - | - | 72.5 | 51.9 | 70.0 | 38.2 | 84.4 |
| w/o NLP | 220k | **56.1** | 64.3 | 65.9 | 74.6 | 52.0 | 69.3 | 39.9 | - |
| w/o Language Modelling | 220k | 52.9 | 64.7 | 66.7 | **74.7** | 52.7 | **70.2** | 39.9 | 83.5 |

Table 4: Ablation study on holding out tasks groups and evaluating on GRIT and MNLI (Williams et al., 2018)

| | NYUv2 | ImageNet | Place365 | VQAv2 | OKVQA | A-OKVQA | VizWizQA | VizWizG | Swig | SNLI-VE | VisComet | Nocaps | COCO | COCO | MRPC | BoolQ | SciTail |
|---|---|---|---|---|---|---|---|---|---|---|---|---|---|---|---|---|---|
| Split | val | val | val | test-dev | test | test | test-dev | test-std | test | val | val | val | val | test | val | val | test |
| Metric | RMSE | Acc. | Acc. | Acc. | Acc. | Acc. | Acc. | IOU | Acc. | Acc. | CIDEr | CIDEr | CIDEr | CIDEr | F1 | Acc | Acc |
| Unified SOTA | UViM | - | - | - | Flamingo | - | Flamingo | - | - | - | - | - | - | - | T5 | PaLM | - |
| | 0.467 | - | - | - | 57.8 | - | 49.8 | - | - | - | - | - | - | - | 92.20 | 92.2 | - |
| UNIFIED-IO_SMALL | - | 42.8 | 38.2 | 57.7 | 31.0 | 24.3 | 42.4 | 35.5 | 17.3 | 76.5 | - | 45.1 | 80.1 | - | 84.9 | 65.9 | 87.4 |
| UNIFIED-IO_BASE | - | 63.3 | 43.2 | 61.8 | 37.8 | 28.5 | 45.8 | 50.0 | 29.7 | 85.6 | - | 66.9 | 104.0 | - | 87.9 | 70.8 | 90.8 |
| UNIFIED-IO_LARGE | - | 71.8 | 50.5 | 67.8 | 42.7 | 33.4 | 47.7 | 54.7 | 40.4 | 86.1 | - | 87.2 | 117.5 | - | 87.5 | 73.1 | 93.1 |
| UNIFIED-IO_XL | 0.475* | 79.1 | 53.2 | 77.9 | 54.0 | 45.2 | 57.4 | 65.0 | 49.8 | 91.1 | 21.2 | 100.0 | 126.8 | 122.3 | 89.2 | 79.7 | 95.7 |
| Single or fine-tuned SOTA | BinsFormer | CoCa | MAE | CoCa | KAT | GPV2 | Flamingo | MAC-Caps | JSL | OFA | SVT | CoCa | - | OFA | Turing NLR | ST-MOE | DeBERTa |
| | 0.330 | 91.00 | 60.3 | 82.3 | 54.4 | 38.1 | 65.7 | 27.3 | 39.6 | 91.0 | 18.3 | 122.4 | - | 145.3 | 93.8 | 92.4 | 97.7 |

*Due to a dataset error we are not able to report results for the SMALL/BASE/LARGE models on NYU2 and report results for an XL model retrained with a slightly different distribution of tasks, see the Appendix A.8 for more discussion.

Table 5: Comparing the jointly trained UNIFIED-IO to specialized and unified fine-tuned SOTA models across Vision, V&L and Language tasks. Benchmark datasets are: NYUv2 (Nathan Silberman & Fergus, 2012), ImageNet (Deng et al., 2009), Places365 (Zhou et al., 2017), VQA 2.0 (Goyal et al., 2017), A-OKVQA (Schwenk et al., 2022), VizWizVQA (Gurari et al., 2018), VizWizG (Chen et al., 2022a), Swig (Pratt et al., 2020), SNLI-VE (Xie et al., 2019), VisComet (Park et al., 2020), Nocaps (Agrawal et al., 2019), COCO Captions (Chen et al., 2015), MRPC (Dolan & Brockett, 2005), BoolQ (Clark et al., 2019), and SciTail (Khot et al., 2018).

## 4.3 RESULTS ON ADDITIONAL TASKS

We report results on 16 additional tasks used in our training setup. For these tasks, we do not expect to get state-of-the-art results since specialized models are usually designed and hyper-parameter tuned for a single task, while we are evaluating a single jointly trained model. We also avoid extensive task-specific tricks like color jittering, horizontal flipping, CIDEr optimization, and label smoothing, which are often responsible for considerable gains in individual task performance. We leave such task-specific tuning for future work. See Table 5 for the results. When possible, we additionally report the best prior result on these tasks from a unified model, meaning a model that is trained in a multi-task setting and a unified architecture (no task-specific head or customizations) with at least three other tasks.

UNIFIED-IO provides strong performance on all these tasks despite being massively multi-tasked. We review more fine-grained results below.

**Depth Estimation.** On depth estimation, UNIFIED-IO achieves 0.475 rmse, which is behind SOTA (Li et al., 2022e) but similar the recently proposed unified model, UViM (Kolesnikov et al., 2022), despite being trained to do far more tasks. More discussion can be found in A.8.

**Image Classification.** UNIFIED-IO achieves 79.1 on ImageNet and 53.2 on Places365, showing the model was able to retain the knowledge of many fine-grained classes despite being massively multi-tasked. Notably, we achieve this without the extensive data augmentations methods typically used by SOTA models (Yu et al., 2022a; He et al., 2022).

**Visual Question Answering.** UNIFIED-IO is competitive with fine-tuned models on VQA (Alayrac et al., 2022; Kamath et al., 2022; Gui et al., 2021), and achieves SOTA results on A-OKVQA. Relative to Flamingo, UNIFIED-IO performs better on VizWiz-QA but worse on OK-VQA.

**Image Captioning.** Despite the lack of CIDEr optimization, UNIFIED-IO is a strong captioning model and generalizes well to nocaps. Since UNIFIED-IO is trained on many captioning datasets, it is likely the use of style tags following Cornia et al. (2021) would offer additional improvement by signaling UNIFIED-IO to specifically generate COCO-style captions during inference.

**NLP tasks.**: UNIFIED-IO achieves respectable results on three NLP tasks but lags behind SOTA models (Smith et al., 2022; Zoph et al., 2022; He et al., 2021). This can partly be attributed to scale. Modern NLP models contain 100 billion+ parameters and with more extensive NLP pre-training.

### 4.4 LIMITATIONS

For object detection, while UNIFIED-IO generally produces accurate outputs (see Appendix A.9), we find the recall is often poor in cluttered images. Prior work (Chen et al., 2022b) has shown this can be overcome with extensive data augmentation techniques, but these methods are not currently integrated into UNIFIED-IO. Our use of a pre-trained VQ-GAN greatly simplifies our training and is surprisingly effective for dense prediction tasks. However, it does mean UNIFIED-IO has limited image generation capabilities (recent works (Yu et al., 2022b) have shown this method can be greatly improved but was not available at the time of development). We also found in a small-scale study that our model does not always understand prompts not in the training data (see Appendix A.6).

## 5 RELATED WORK

Constructing models that can learn to solve many different tasks has been of long-standing interest to researchers. A traditional approach to this problem is to build models with task-specialized heads on top of shared backbones (He et al., 2017; Liu et al., 2019; Lu et al., 2020). However, this requires manually designing a specialized head for each task and potentially limits transfer across tasks. An alternative is to build *unified* models – models that can complete many different tasks without task-specialized components. In NLP, this approach has achieved a great deal of success through the use of pre-trained generative models (Raffel et al., 2020; Brown et al., 2020; Chowdhery et al., 2022).

Inspired by this success, there has been a recent trend to build unified models that can be additionally applied to tasks with visual or structured inputs and outputs. Many models have been proposed for tasks with text and/or image input and text output (Cho et al., 2021; Wang et al., 2022d; Li et al., 2022b; Wang et al., 2021; Kaiser et al., 2017; Sun et al., 2022; Chen et al., 2022d; Wang et al., 2022c). However, these models can not produce any structured or visual output.

More recent unified models can additionally support image locations, which allows tasks like object detection or region captioning. This can be done by using bounding boxes proposed by an object detector (Cho et al., 2021; Kamath et al., 2022) or including a bounding box output head (Gupta et al., 2022a; Dou et al., 2022; Chen et al., 2022c; Kamath et al., 2021; Li et al., 2022d). Alternatively, image locations can be encoded as special tokens in the input/output text (Yang et al., 2021; Yao et al., 2022; Zhu et al., 2022) following Chen et al. (2022b). UNIFIED-IO follows this design, but applies it to a wider set of tasks than previous works, including key-point estimation, image in-painting, and region captioning.

Concurrent to our work, OFA (Wang et al., 2022b) proposes a similar approach that also supports image locations and text-to-image synthesis. However, OFA does not support dense labeling tasks such as depth estimation, segmentation, and surface normal estimation. Other closely related models include UViM (Kolesnikov et al., 2022) which generates a discrete guiding code for a D-VAE to build an autoregressive model for panoptic segmentation, depth prediction, and colorization; Pix2Seq v2 (Chen et al., 2022c) which extends Pix2Seq to segmentation, keypoint estimation, and image captioning; Visual Prompting (Bar et al., 2022) adapts the pre-trained visual model to novel downstream tasks by image inpainting. UNIFIED-IO covers all these tasks, and focuses on multi-tasking rather then task-specific fine-tuning. Additional discussions are presented in Appendix A.10.

## 6 CONCLUSION

We have presented UNIFIED-IO, a unified architecture that supports a large variety of computer vision and NLP tasks with diverse inputs and outputs, including images, continuous maps, binary masks, segmentation masks, text, bounding boxes, and keypoints. This unification is made possible by homogenizing each of these modalities into a sequence of discrete tokens. The 2.9B parameter UNIFIED-IO XL model is jointly trained on 90+ datasets, is the first model to perform all 7 tasks on the GRIT benchmark and obtains impressive results across 16 other vision and NLP benchmarks, with no benchmark fine-tuning or task-specific modifications.

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

# A  APPENDIX

## A.1  TASKS DETAILS

UNIFIED-IO is jointly trained on a large and diverse set of vision, language and vision & language tasks. In this section, we describe these tasks in detail and show the prompts we use during training and inference (text on the left of example cards). We also provide qualitative examples of both the ground truth and the predictions made by UNIFIED-IO.

### A.1.1  IMAGE SYNTHESIS TASKS

**Image Synthesis from Text.** This task requires generating an image that matches a sentence. Training data comes from 4 captioning datasets: COCO Caption (Chen et al., 2015), Conceptual Captions 3M and 12M (Changpinyo et al., 2021), and RedCaps (Desai et al., 2021) as well datasets used for image classification using the object class as the input caption. Specialized image generation models like DALL·E 2 (Ramesh et al., 2022) use an order of magnitude more data, but we limit our sources to these sets for training efficiency.

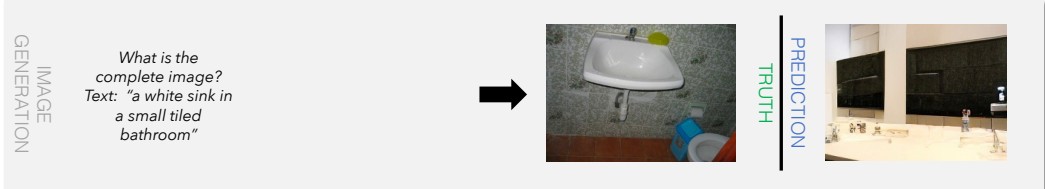

**Image Inpainting.** This task requires filling in a region of an image with a target object. Training data for this task is built from object bounding box annotations from Open Images (Kuznetsova et al., 2020), Visual Genome (Krishna et al., 2017) and COCO (Lin et al., 2014). For each object, the input image becomes the source image with the object's bounding box blanked out. The input prompt provides the bounding box's location and the target category. The target output is the original image.

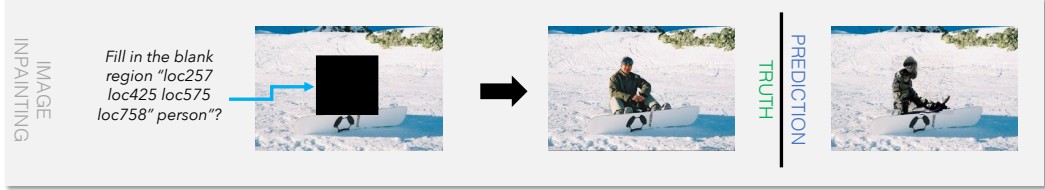

**Image Synthesis from Segmentation.** This task involves generating an image that matches an input semantic segmentation, i.e., a set of class labels for some or all of the pixels in the image. UNIFIED-IO is trained for this task using segmentation annotations from COCO (Lin et al., 2014), Open Images (Kuznetsova et al., 2020), and LVIS (Gupta et al., 2019) as input. Following the method from Section 3.1 the segmentation input is converted into a RGB image paired with a prompt listing the color-to-class mapping, and the target output is the source image.

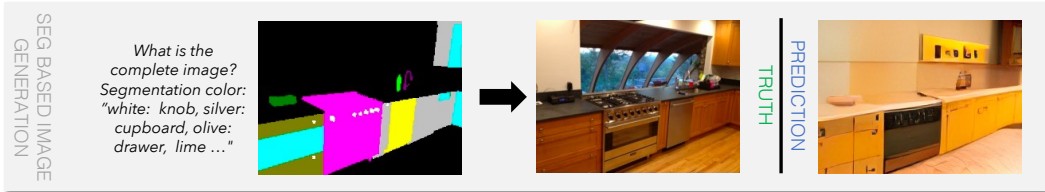

### A.1.2  SPARSE LABELLING TASKS

**Object Detection.** UNIFIED-IO is trained on object detection annotations from Visual Genome, Open Images, and COCO. For this task the input is a static prompt and an image, and the output

text includes the bounding boxes and class names of all objects in the image. We randomize the order of the output objects during training, but for simplicity leave integrating more complex data-augmentation techniques (Chen et al., 2022b) to future work.

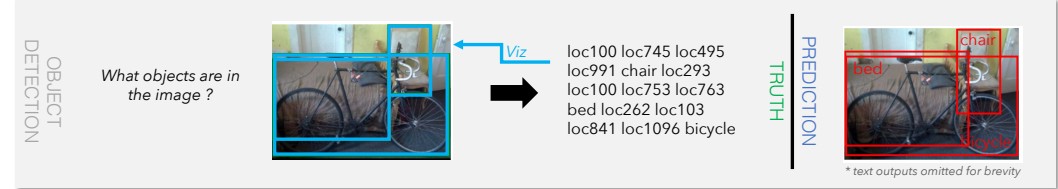

**Object Localization.** Object localization requires returning bounding boxes around all objects of a given category. Training data is derived from our object detection training data by constructing a training example from each category of objects present in an image. The input is then the image, a prompt specifying the target class, and the output is a list of all boxes that contain an instance of that class. The class for each box (which is always the class specified in the prompt) is included in the output for the sake of keeping the output format consistent with the object detection output. Object localization can use input categories which are not present in the image. To handle this, we construct negative samples by randomly selecting categories not present in the image to use as input, in which case the output is an empty sequence.

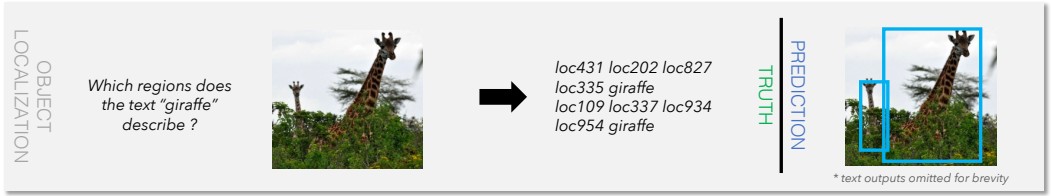

**Referring Expression Comprehension.** The task requires the model to localize an image region described by a natural language expression. The annotation is similar to Object Localization, except that the target is specified with natural language expression instead of class name. Datasets for this task include RefCOCO (Kazemzadeh et al., 2014), RefCOCO+ (Kazemzadeh et al., 2014) and RefCOCOg (Mao et al., 2016).

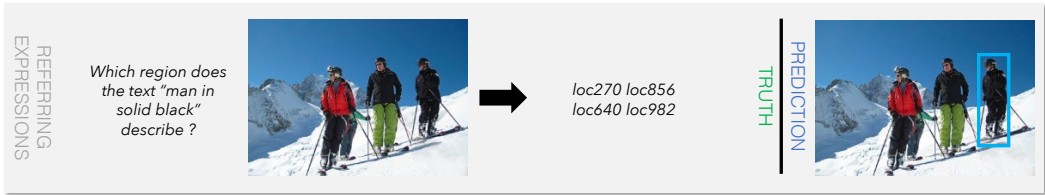

**Keypoint Estimation.** Keypoint estimation requires returning the location of 17 keypoints on a human body (e.g., eyes, nose, feet, etc.) for each person in an image. While it is possible to perform this task in one pass by listing the keypoints of all people in the image in a single output sequence, this can result in an extremely long output sequence, so UNIFIED-IO uses a multi-step approach instead. To do this UNIFIED-IO is trained to complete the subtask of detecting the keypoints for single a person in a given region. For this subtask, the input prompt specifies the target region and and the output is a list of 17 points (a pair of locations tokens for the $x$ and $y$ coordinates) along with a visibility labels (1 for not visible, 2 for partly visible, 3 for fully visible). Non-visible points are preceded by two copies of a new special tokens that indicate there are no valid coordinates. The keypoint metric does not award points for correctly identifying non-visible points, so during inference we mask that special token so the model makes a best-effort guess for the coordinates of every single point. Training data for this subtask comes from COCO human pose data (Lin et al., 2014) with the ground-truth person regions as input. During inference we locate person regions using the object localization prompt, then apply UNIFIED-IO again to find keypoints for each detected region.

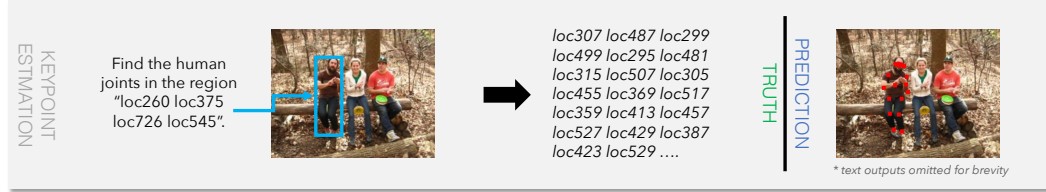

### A.1.3 DENSE LABELLING TASKS

**Object Segmentation.** Object segmentation requires finding the binary segmentation mask of each instance of a particular category in an image. The input is an image and a prompt that includes the target class, while the output is an RGB image with black background and instances of that class filled in with unique colors following the method in Section 3.1. The output image is resized to match the input image if needed using a nearest-neighbor resizing method, and binary masks are built from each unique color. In practice the output image from UNIFIED-IO can have slightly non-uniform colors or extraneous background pixels, likely due to limitation in what the D-VAE can decode/encode, so the output pixels are clustered by color and and connected components of less than 8 pixels are removed to build cleaned instance masks. Segmentation annotations come from Open Images LVIS, and COCO.

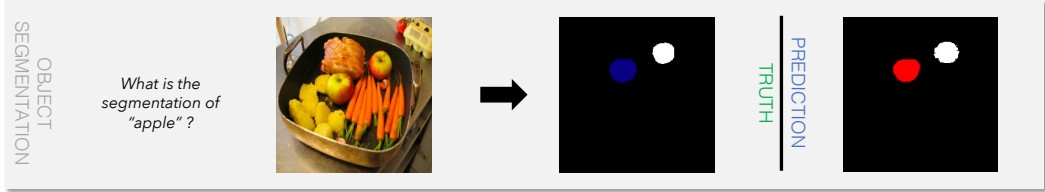

**Depth Estimation.** Depth estimation requires assigning each pixel in an image a depth value. This task uses a static prompt as input, and the output is a grayscale image representing the normalized depth at each pixel. The generated output image is reiszed to the same size as the input image and then pixel values are rescaled to the maximum depth in the training to get an output depth map. Training data comes from the NYU Depth Dataset V2 (Nathan Silberman & Fergus, 2012).

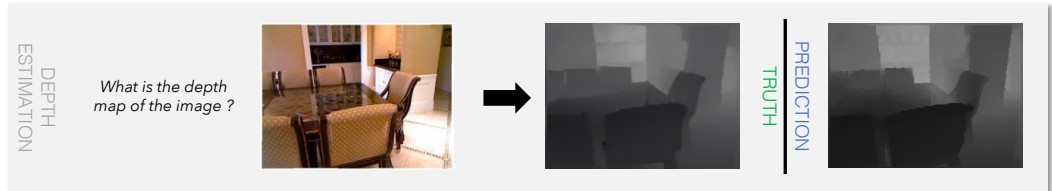

**Surface Normal Estimation.** UNIFIED-IO is trained on FrameNet (Huang et al., 2019a) and BlendedMVS (Yao et al., 2020) surface normal estimation datasets. For this task the input is a static prompt and an image and the output is an RGB representation of the $x/y/z$ orientation of the surface at each pixel. The generated output image is resized to match the input image and converted back to $x/y/z$ orientations to produce the final output.

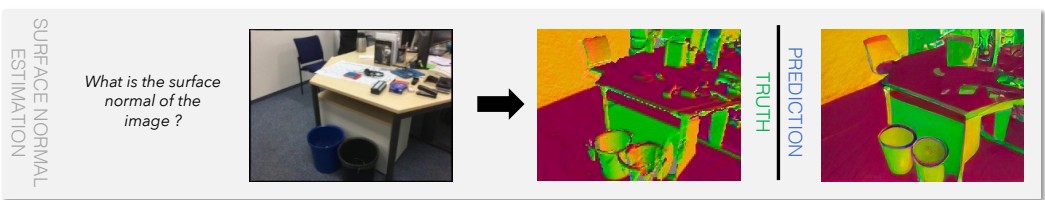

### A.1.4  IMAGE CLASSIFICATION TASKS

**Image Classification.**  UNIFIED-IO is trained on 6 image classification datasets: ImageNet 2012 (Deng et al., 2009), ImageNet21k (Ridnik et al., 2021), Places365 (Zhou et al., 2017), Sun397 (Xiao et al., 2010), iNaturalist (Van Horn et al., 2018) and Caltech birds 2011 (Welinder et al., 2010). For this task the input is an image and a static prompt, and the output is a class name. During inference we compute the log-probability of each class label in the dataset being evaluated and return the highest scoring one. This ensures UNIFIED-IO does not return a category from a different categorization dataset that is a synonym or hypernym of the correct label.

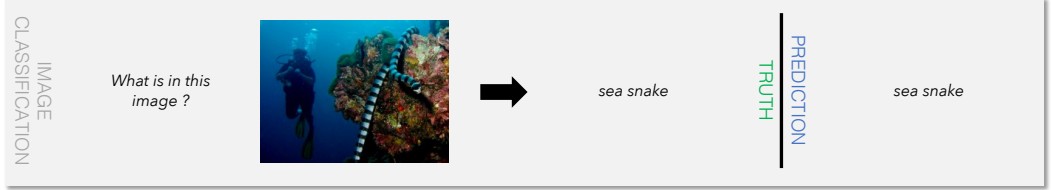

**Object Categorization.**  This task identifies which label, from a given set, best corresponds to an image region defined by an input image and bounding box. The input is the image, a prompt specifying the image region and the output is the target class name. We convert object detection annotations from Visual Genome, Open Images, and COCO for this task. Inference is constrained to return a valid label for the target label set just as with image classification.

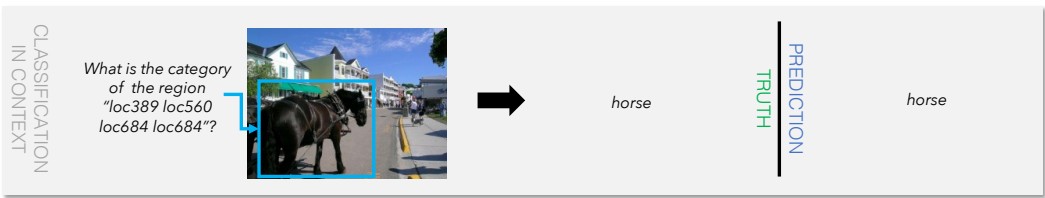

### A.1.5  IMAGE CAPTIONING TASKS

**Image Captioning.** Image captioning data comes from the same manually annotated and unsupervised sources used for Image Generation. In this case the inputs and output are reversed, the input is an image and the static prompt, and the output is a caption that matches the image.

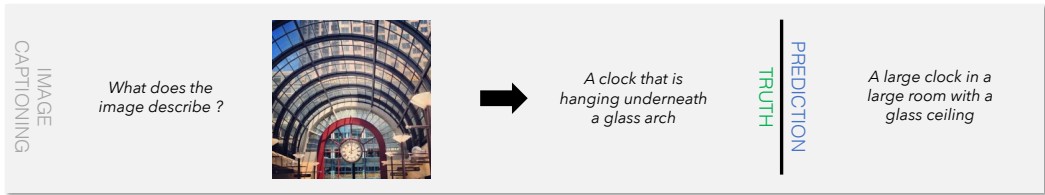

**Region Captioning.** Region captioning tasks a model with generating a caption that describes a specific region in the image. Our format for this task is identical to Image Captioning except the region is included in the input prompt. Visual Genome (Krishna et al., 2017) is used for the training data.

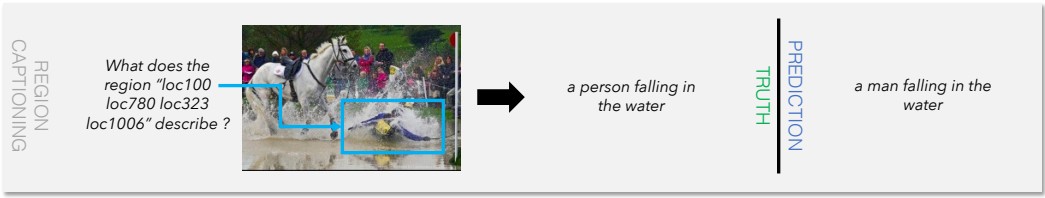

### A.1.6 VISION & LANGUAGE TASKS

**Visual Question Answering.** UNIFIED-IO is trained on a collection of VQA datasets including VQA 2.0 (Goyal et al., 2017), Visual Genome, VizWizVQA (Gurari et al., 2018), OKVQA (Marino et al., 2019) and A-OKVQA (Schwenk et al., 2022). For VQA, the question is used as the prompt, and the output is the answer text. For VQA, it is common to constrain the model to predict an answer from a fixed last of common VQA answers (Wang et al., 2022b;d) during inference, but we avoid doing this since we find it does not benefit UNIFIED-IO in practice.

We additionally convert data from several other datasets in a VQA format, including imSitu (Yatskar et al., 2016), where we treat predicting the verb and then the related slots as separate VQA questions, VisualCOMMET (Park et al., 2020) where we convert the before/after/intent into questions by converting the input regions into location tokens, SNLI-VE (Xie et al., 2019) where we integrate the entailed text into an input question, and VCR (Zellers et al., 2019a) where we again integrate the input regions into the prompt by encoding them with location tokens and integrate the rationales into the target text for the answer justification task.

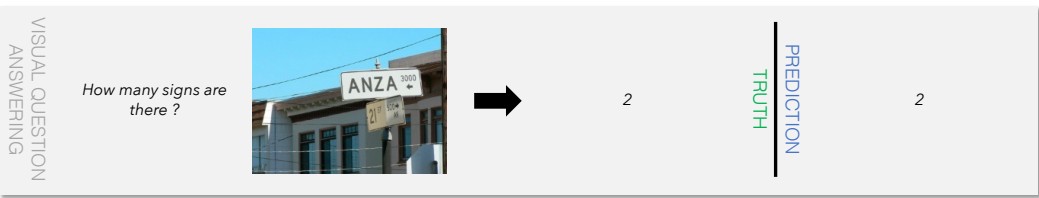

**Answer-Grounded Visual Question Answering.** This task requires both answering a question and returning a binary mask specifying the region of the image used to answer the question. The format for this task follows the one for VQA except that a binary mask is also used as an additional output. Training data comes from VizWiz-VQA (Chen et al., 2022a), a dataset designed to train models that could benefit people with visual impairments.

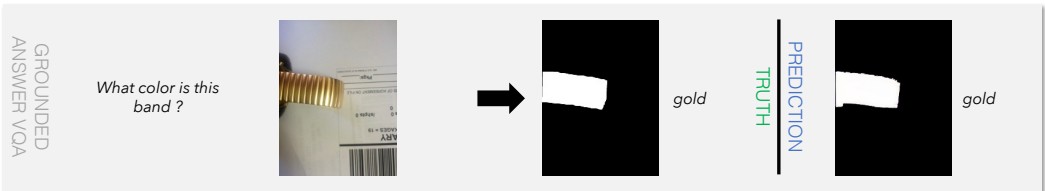

**Relationship Detection.** This task requires predicating a relationship between a pair of objects which are grounded by bounding boxes. The prompt contains both the object regions, and the output is the predicted predicate. There are 2 datasets in this tasks: Visual Genome (Krishna et al., 2017) and Open Images (Kuznetsova et al., 2020).

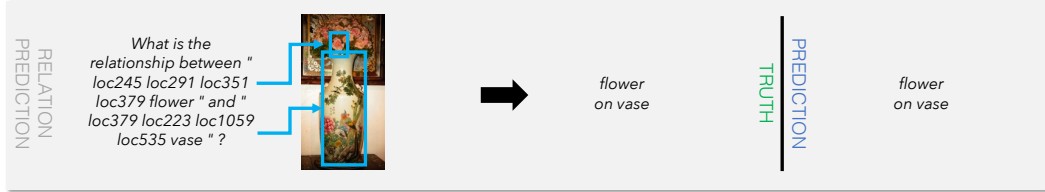

### A.1.7 NATURAL LANGUAGE PROCESSING TASKS

**Question Answering.** Following prior work in natural language processing (Raffel et al., 2020), QA tasks are formatted by placing both the question and any text context (e.g., an paragraph containing the answer) into the prompt and training the model to generate the text answer. UNIFIED-IO is trained on several QA datasets including SQuAD 2.0 (Rajpurkar et al., 2016), other training datasets from the MRQA (Fisch et al., 2019) shared tasks (Trischler et al., 2017; Joshi et al., 2017; Dunn et al., 2017; Yang et al., 2018; Kwiatkowski et al., 2019), QA datasets from SuperGLUE (Wang

et al., 2019; Clark et al., 2019; Khashabi et al., 2018; Roemmele et al., 2011), Cosmos QA (Huang et al., 2019b), OpenBookQA (Mihaylov et al., 2018), and HellaSwag (Zellers et al., 2019b). If the text context is longer then our maximum sequence length we use a sliding-window approach following Devlin et al. (2019) which exposes the model to different windows of text from the context and returns the highest-confidence answer.

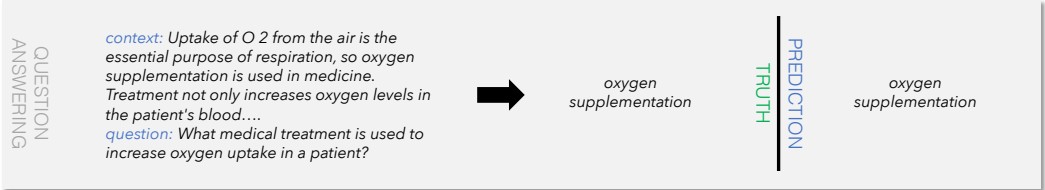

**Text Classification.** Also following past work (Raffel et al., 2020), text classification tasks are formatted by placing the input sentences and a query in the prompt and training the model to generate the target class. Datasets include tasks from GLUE and SuperGLUE (Wang et al., 2018; 2019; Warstadt et al., 2018; Socher et al., 2013; Dolan & Brockett, 2005; Iyer et al., 2017; Cer et al., 2017; Williams et al., 2018; Dagan et al., 2005; Bar-Haim et al., 2006; Giampiccolo et al., 2007; Bentivogli et al., 2009; Levesque et al., 2012; Williams et al., 2018; De Marneff et al., 2019; Pilehvar & os'e Camacho-Collados, 2018), as well as SNLI (Bowman et al., 2015), SciTail (Khot et al., 2018), IMDB Reviews (Maas et al., 2011), and PAWS (Zhang et al., 2019).

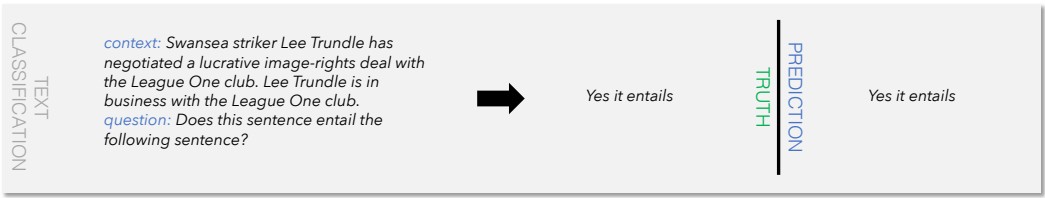

**Text Summarization.** Text summarization is done again by providing the input paragraph and a prompt as input and generating a summary as output. We use the Gigaword dataset (Graff et al., 2003; Rush et al., 2015) for training data.

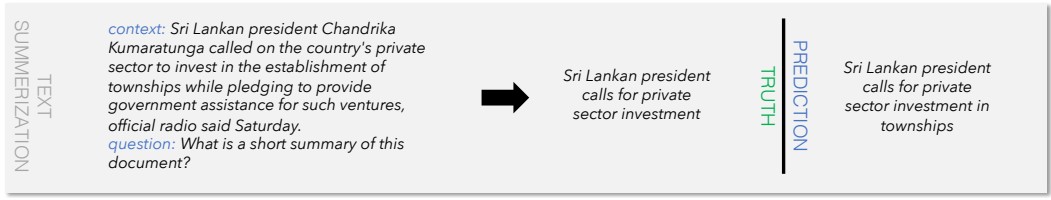

### A.1.8 LANGUAGE MODELING TASKS

**Mask Language Modeling.** Following T5 (Raffel et al., 2020), the mask language modelling objective randomly samples and then drops out 15% of tokens in the input sequence. All consecutive spans of dropped-out tokens are replaced by a single sentinel token. The target is to recover the dropped tokens given the sentinel token. We use C4 (Raffel et al., 2020) and Wikipedia (Foundation) datasets.

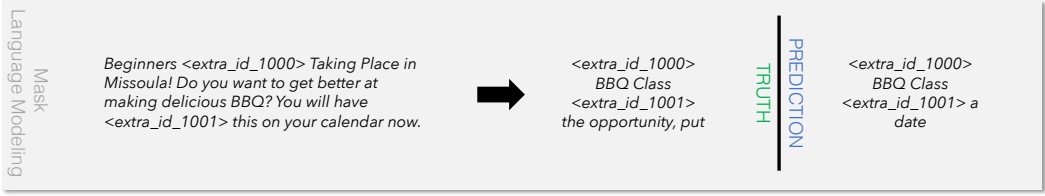

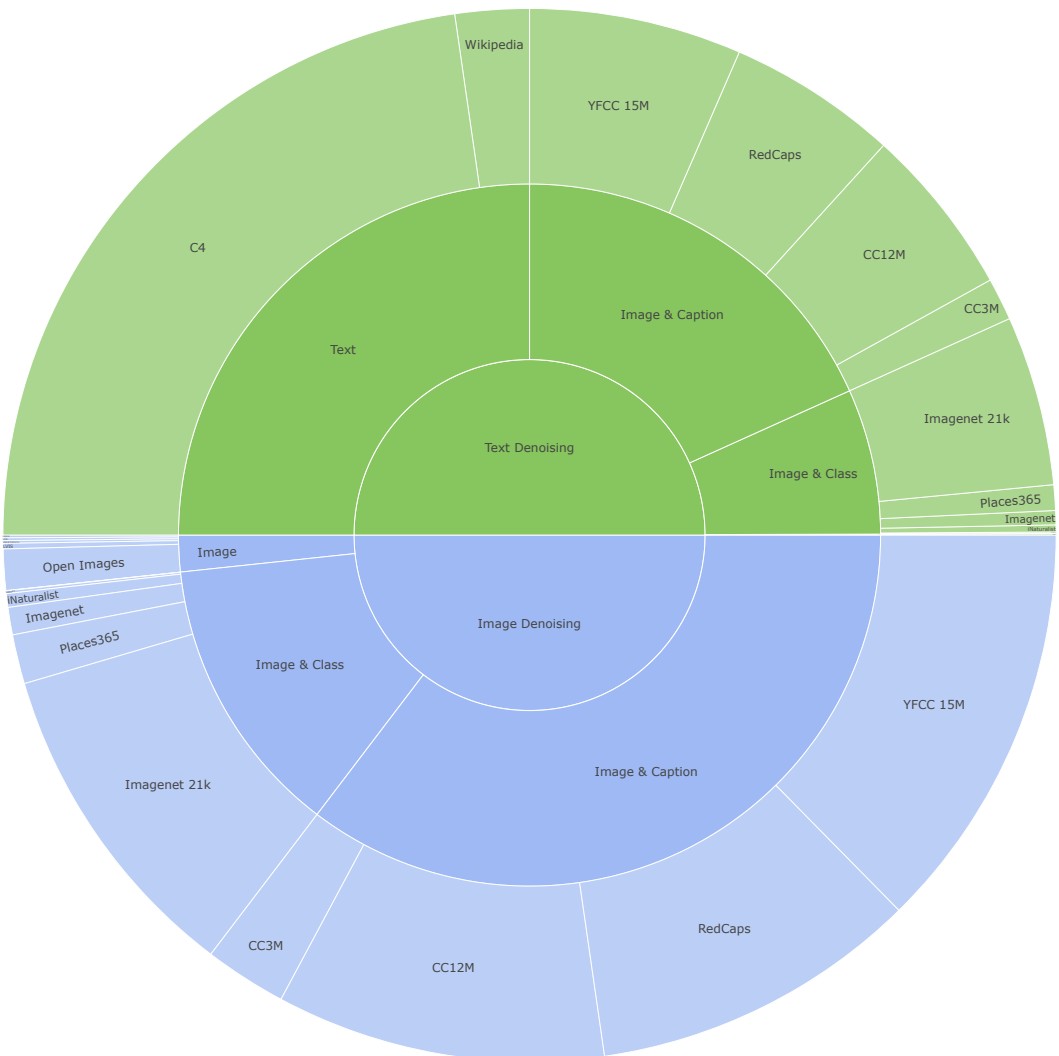

Figure 3: Pre-training objectives (inner circle), annotation types (middle circle), and datasets (outer circle) used in pre-training of UNIFIED-IO. Sizes correspond to the sampling rate in the training distribution. Best viewed in color.

## A.2 PRE-TRAINING DATA DISTRIBUTION

Figure 3 shows a visualization of pre-training data distribution used by UNIFIED-IO. As discussed in Section 3.3, we equally sample data with the text denoising and image denoising objective (inner circle of Figure 3). For text denoising, half of the samples are from pure text data, *i.e.* C4 and Wikipedia. The other half is constructed from image and class, such as Imagenet21k (Ridnik et al., 2021) or image and caption, such as YFCC15M (Radford et al., 2021). For image denoising, we use the text information when class and caption are present in the data source and sample the dataset proportional to the dataset size. For both text and image denoising, we randomly drop both modalities 10% of the time if both text and image as inputs.

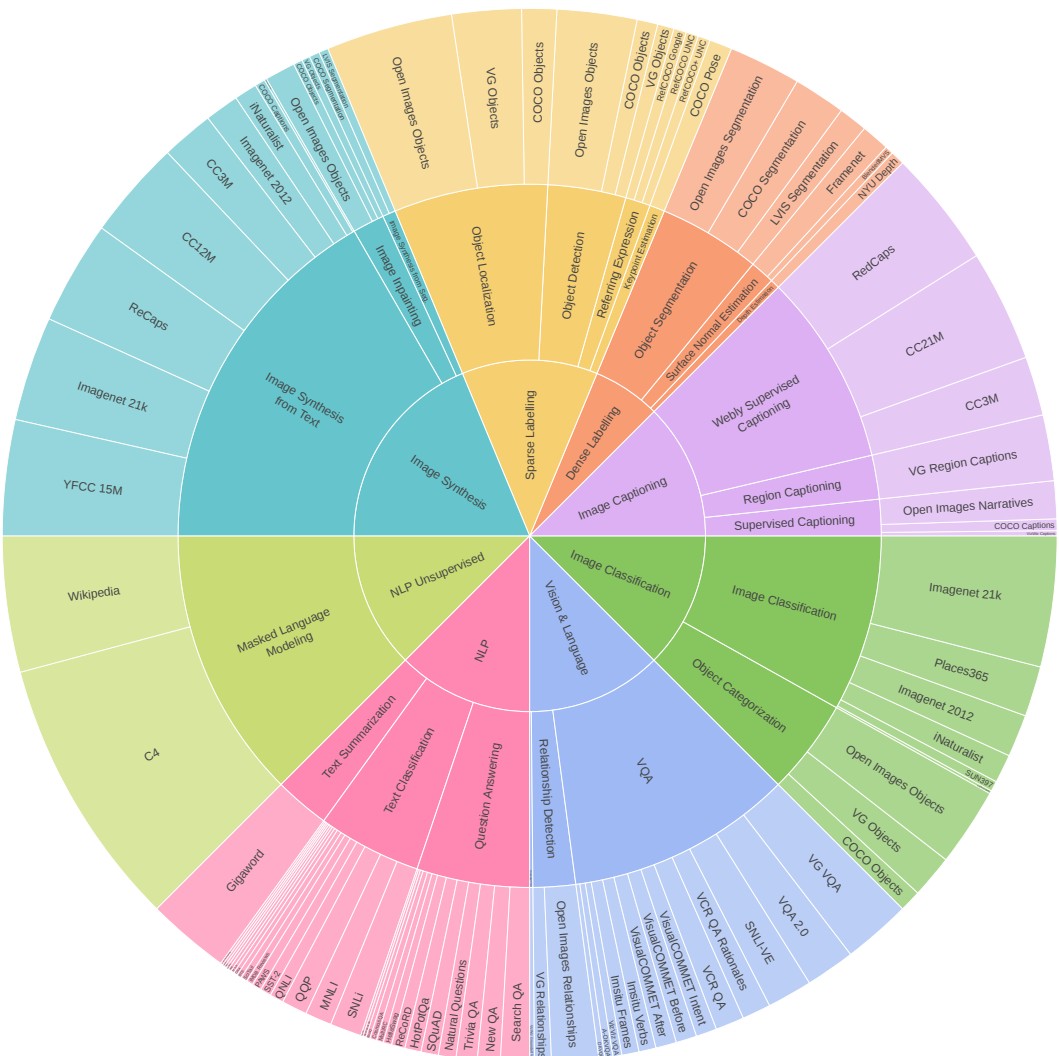

Figure 4: Task groups (inner circle), tasks (middle circle), and datasets (outer circle) used in multi-task training of UNIFIED-IO. Sizes correspond to the sampling rate in the training distribution. Best viewed in color.

## A.3  MULTI-TASKING DATA DISTRIBUTION

Figure 4 shows a visualization of the multi-task training distribution used by UNIFIED-IO from Table 1. As discussed in Section 3.3, we equally sample each group (1/8) except image synthesis (3/16) and dense labeling (1/16) since dense labeling has a much smaller sample size compared to image synthesis. We sample tasks and datasets (middle and outer circle) with a temperature-scaled mixing strategy to make sure the model is sufficiently exposed to underrepresented tasks. We raise each task's mixing rate to the power of $1/T$ and then renormalize the rates so that they sum to 1. Following Raffel et al. (2020), we use $T = 2$ in our experiments.

Due to the large variance in dataset size, some of the tasks are rarely sampled. For example, the depth estimation task only has the NYU Depth dataset source (Nathan Silberman & Fergus, 2012) and thus the sampling rate is only 0.43%. However, the model still works well for depth estimation tasks, even outperforming concurrent work (Kolesnikov et al., 2022) (0.385 vs. 0.467 RMSE). We suspect the large model capacity and masked image denoising pre-training improve the performance. Similarly, Grounding VQA (Chen et al., 2022a) has 0.15% sample rate, but the model can still achieve state-of-the-art performance on this task partly because it is trained on many related datasets for VQA and segmentation.

## A.4 IMPLEMENTATION DETAILS

The total vocabulary size is 49536, with 32152 language tokens, 1000 location tokens, and 16384 vision tokens. We use the imagenet pretrained VQ-GAN checkpoints with 16384 tokens and $f = 16$[1]. Please refer (Esser et al., 2021) During training, we random sub-sample 128 image patches for pre-training state and 256 image patches (out of 576) for multi-task stage. We do not use dropout. Adafactor (Shazeer & Stern, 2018) optimizer is used to save memory. We use a learning rate of $10^{-2}$ for the first 10,000 steps and then decay at a rate of $1/\sqrt{k}$. We train with $\beta_1 = 0.9$ and $\beta_2 = 1.0 - k^{-0.8}$, where $k$ is the step number. We use global norm gradient clipping with 1.0 and find this is crucial to stabilized XL training. We train the Small, Base and Large with a batch size of 204 8 and XL with batch size of 1024 due to memory consideration. 4-way in-layer parallelism and 128-way data parallelism were used to scale the 3B model training. For all models, we train $1000k$ steps – $500k$ for pre-training and multi-task training respectively.

## A.5 EVALUATION ON THE SAME CONCEPT AND NEW CONCEPT

| | restricted | params (M) | Categorization | | Localization | | VQA | | Refexp | | Segmentation | | Keypoint | | Normal | |
|---|---|---|---|---|---|---|---|---|---|---|---|---|---|---|---|---|
| | | | same | new | same | new | same | new | same | new | same | new | same | new | same | new |
| 0 NLL-AngMF | ✓ | 72 | - | - | - | - | - | - | - | - | - | - | - | - | **50.7** | - |
| 1 Mask R-CNN | ✓ | 58 | - | - | 51.9 | 40.8 | - | - | - | - | 44.9 | 0.3 | **70.9** | - | - | - |
| 2 GPV-1 | ✓ | 236 | 58.7 | 0.8 | 48.3 | 37.8 | 58.4 | 74.0 | 29.7 | 23.1 | - | - | - | - | - | - |
| 3 CLIP | | 302 | 49.1 | 46.7 | - | - | - | - | - | - | - | - | - | - | - | - |
| 4 OFA$_{\text{LARGE}}$ | | 473 | 28.9 | 15.8 | - | - | 74.9 | 88.6 | 63.4 | 58.5 | - | - | - | - | - | - |
| 5 GPV-2 | | 370 | **85.0** | 13.5 | 54.6 | 54.2 | 69.8 | 81.7 | 57.8 | 48.3 | - | - | - | - | - | - |
| 6 UNIFIED-IO$_{\text{SMALL}}$ | | 71 | 52.9 | 31.9 | 47.5 | 61.5 | 59.0 | 72.5 | 54.2 | 45.7 | 37.4 | 48.5 | 46.6 | - | 33.6 | - |
| 7 UNIFIED-IO$_{\text{BASE}}$ | | 241 | 60.3 | 47.5 | 57.9 | 68.4 | 68.0 | 81.8 | 72.5 | 62.2 | 45.8 | 57.2 | 60.2 | - | 37.7 | - |
| 8 UNIFIED-IO$_{\text{LARGE}}$ | | 776 | 63.0 | 52.7 | 63.3 | 70.9 | 72.1 | 84.3 | 79.2 | 66.3 | 50.4 | 62.2 | 67.7 | - | 40.3 | - |
| 9 UNIFIED-IO$_{\text{XL}}$ | | 2925 | 66.1 | **60.1** | **65.6** | **74.4** | 78.6 | **90.2** | **83.5** | 72.4 | **53.0** | **64.2** | 68.2 | - | 45.1 | - |

Table 6: Generalization to new concepts on the GRIT ablation set.

GRIT provides a breakdown of metrics into two groups: *same* for samples that only contain concepts seen in the primary training data (a set of common datasets like COCO, ImageNet, and Visual Genome), and *new* for samples containing at least one concept unseen in primary training data. Table 6 shows results for UNIFIED-IO and other leaderboard entries for the ablation set, divided into the same and new concepts.

UNIFIED-IO$_{\text{XL}}$ shows little degradation in performance between *same* and *new*, compared to competing entries. On some tasks UNIFIED-IO is even able to outperform on the *new* split compared to the *same*. This indicates that the volume of training data used to train UNIFIED-IO has a broad coverage of concepts, and provides almost as effective a level of supervision as provided by large standard vision datasets like COCO. Furthermore, since UNIFIED-IO is a uniquely unified architecture with no task-specific parameters, it is very likely able to effectively transfer knowledge across different tasks.

In comparison to Mask-RCNN (row 1), GRIT metrics show UNIFIED-IO (row 14) is better by a large margin on *new* concepts, i.e., non-COCO examples (74.4 vs 40.8 for localization and 64.2 vs 0.3 on segmentation), but is still superior on the COCO-like examples (65.6 vs 51.9 for localization and 53.0 vs 44.9 on segmentation). UNIFIED-IO is also able to beat GPV-2 (row 5) on *new* concepts by large margins across all 4 tasks supported by GPV-2 even though GPV-2 is exposed to these concepts via webly supervised data and is designed to transfer concept knowledge across skills.

## A.6 PROMPT GENERALIZATION CASE STUDY

To better understand how different prompts affect UNIFIED-IO, we do a case study on referring expressions. In particular, we re-evaluate UNIFIED-IO on the GRIT referring expression ablation set while replacing the prompt used during training (first row in the table) with a paraphrase (following rows). Results are shown in Table 7.

---

[1]https://github.com/CompVis/taming-transformers

|   | Prompt | Refexp Score |
|---|--------|--------------|
| 0 | Which region does the text " REFEXP " describe ? | 78.9 |
| 1 | Which region does the text "REFEXP" describe? | 76.7 |
| 2 | Which region matches the text " REFEXP " ? | 77.4 |
| 3 | Locate the " REFEXP " . | 65.6 |
| 4 | Which region can be described as " REFEXP " ? | 64.8 |
| 5 | Locate the region described by " REFEXP " . | 43.2 |
| 6 | Where is the " REFEXP " ? | 41.5 |
| 7 | Where is the "REFEXP"? | 0.1 |

Table 7: Case study on GRIT referring expressions using different prompts. The first prompt is the one used during training, the others are paraphrases. REFEXP is replaced by the referring expression text of individual examples during evaluation.

Overall, we find that the model has some capacity to generalize to paraphrases of the prompt (*e.g.*, row 3 works reasonably well despite using completely different words), but there are paraphrases that result in a very significant performance decrease (*e.g.* rows 5, 6, and 8). We also find removing the spaces around the punctuation sometimes results in minor regressions (row 0 vs row 1) and sometimes in sharply reduced performance (row 6 vs row 7), showing UNIFIED-IO can be sensitive to formatting details. We hypothesize that this is caused by the SentencePiece tokenizer changing the tokenization of the referring expression if the quotes are not separated from it by spaces. Building multi-task models that can generalize to different prompts, and ideally to prompts for completely new tasks, is an exciting avenue for future work.

## A.7 CROSS TASK GENERALIZATION CASE STUDY

Qualitative examples for UNIFIED-IO when applied to two out-of-domain settings, surface normal detection on COCO images and animal pose estimation, are included in Figure 5 with the other qualitative examples. For surface normal detection, we find that the model produces plausible images even for objects like cats or humans that are not in the surface normal training data. However, other scenes, such as outdoor scenes (Figure 5 bottom right), are less coherent.

Despite not being trained on animal pose estimation, UNIFIED-IO is able to sometimes find animal keypoints. For animals standing or crouching on two legs, the keypoints are reasonably accurate (first two images), however for animals standing on four legs the model will find leg and eye points but then guess arm positions that would make sense for a person instead of attaching points to the other legs. While this hints that the model was able to combine skills learned from human pose estimation data with the knowledge of animals learned from other tasks, it also shows that more work is needed to fully realize this potential.

## A.8 NYUv2 RESULTS

The first version of UNIFIED-IO multi-tasking data distribution contains two sources of depth dataset. `nyu_depth_v2` from Tensorflow Dataset[2] and a pre-processed version from `sparse-to-dense.pytorch`[3]. Since the original NYUv2 dataset has a lot of 0-distance regions (holes) which can be problematic for sequence training, we included the latter source because it replaces the 0-distance holes with approximations. The second version of UNIFIED-IO model we trained for code release is trained on an updated multi-tasking data distribution that only contains the Tensorflow source of the NYUv2 dataset, and we find a significant drop in performance for XL model 0.475 *vs*. 0.385 while other tasks maintain similar performance. We suspect the reason is `sparse-to-dense.pytorch` has a different split and contaminates the training data for NYUv2 evaluation. Our final result is a little worse compared to UViM 0.475 *vs*. 0.467.

---

[2]https://www.tensorflow.org/datasets/catalog/nyu_depth_v2
[3]http://datasets.lids.mit.edu/sparse-to-dense/data/nyudepthv2.tar.gz

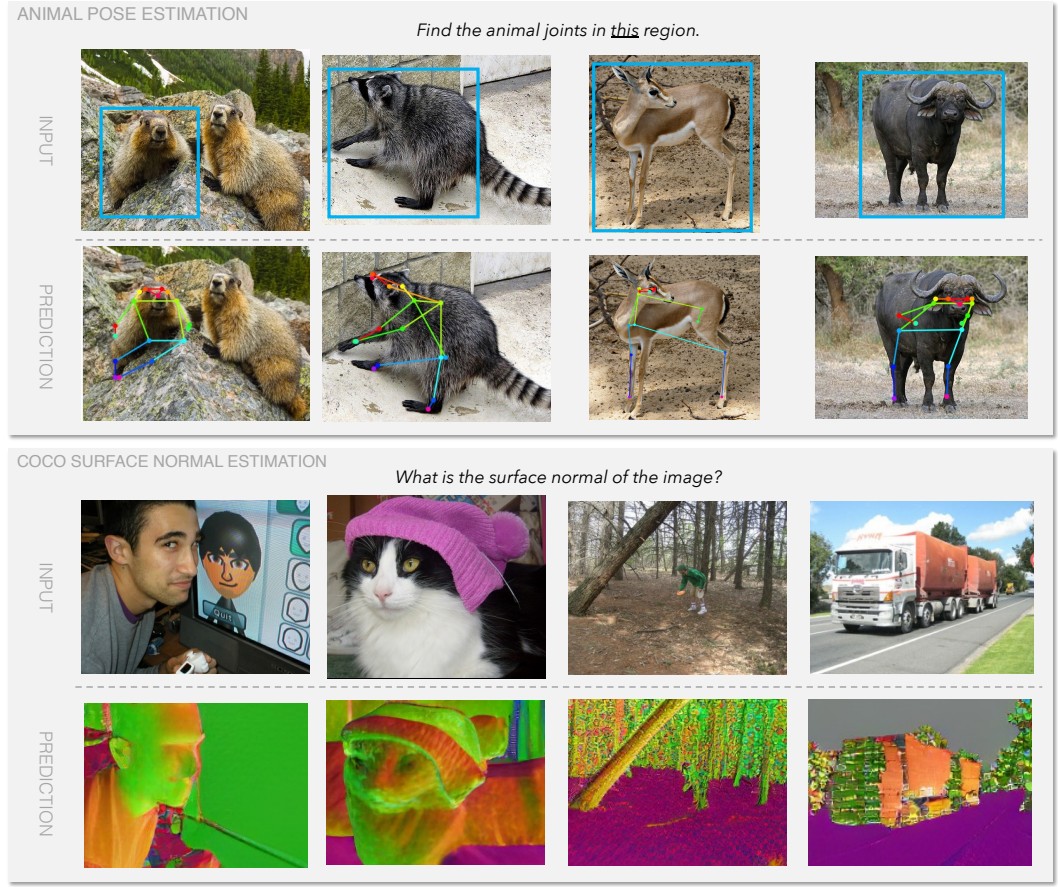

Figure 5: Qualitative examples on out-of-domain inputs. We directly evaluate UNIFIED-IO_XL on animal pose estimation and surface normal of human or outdoor scenes.

## A.9 QUALITATIVE EXAMPLES

Here we present qualitative examples of predictions from UNIFIED-IO for all training tasks. For brevity, if prompts are identical for each example we only show the prompt once, and if the prompt follow the same template for each example we show the template with parts that would be substituted with different words or location tokens underlined, and then show just the substitution with individual examples.

## A.10 OTHER RELATED WORK

**Other Modalities.** Multi-modal models for video (Li et al., 2022c;a; Wang et al., 2022a; Alayrac et al., 2022; Zellers et al., 2021; Yu et al., 2022a), audio (Zellers et al., 2022; Jaegle et al., 2022), and other modalities including game-playing and robot controlling (Reed et al., 2022; Jaegle et al., 2022; Liang et al., 2022) have also been studied. Integrating these modalities is an important line of research, however existing models often do not even support sparse structured output and do not support dense structured outputs, so they do not meet our objective of supporting classic vision tasks.

**Vision & Language Pre-Training.** Vision and language pre-training has become standard practice for multi-modal models, including both unified models and non-unified models that require task-specific heads to be trained from scratch during fine-tuning. Many initial pre-training strategies were inspired by BERT (Devlin et al., 2019) and included masked-language-modeling, image-text-matching, or mask-region-modeling objectives, often supplemented with objectives using the predictions of a strong object detector model (e.g, VILBERT (Lu et al., 2019), LXMERT (Tan & Bansal, 2019), VisualBERT (Li et al., 2019)). More recently contrastive-image-text losses (Radford

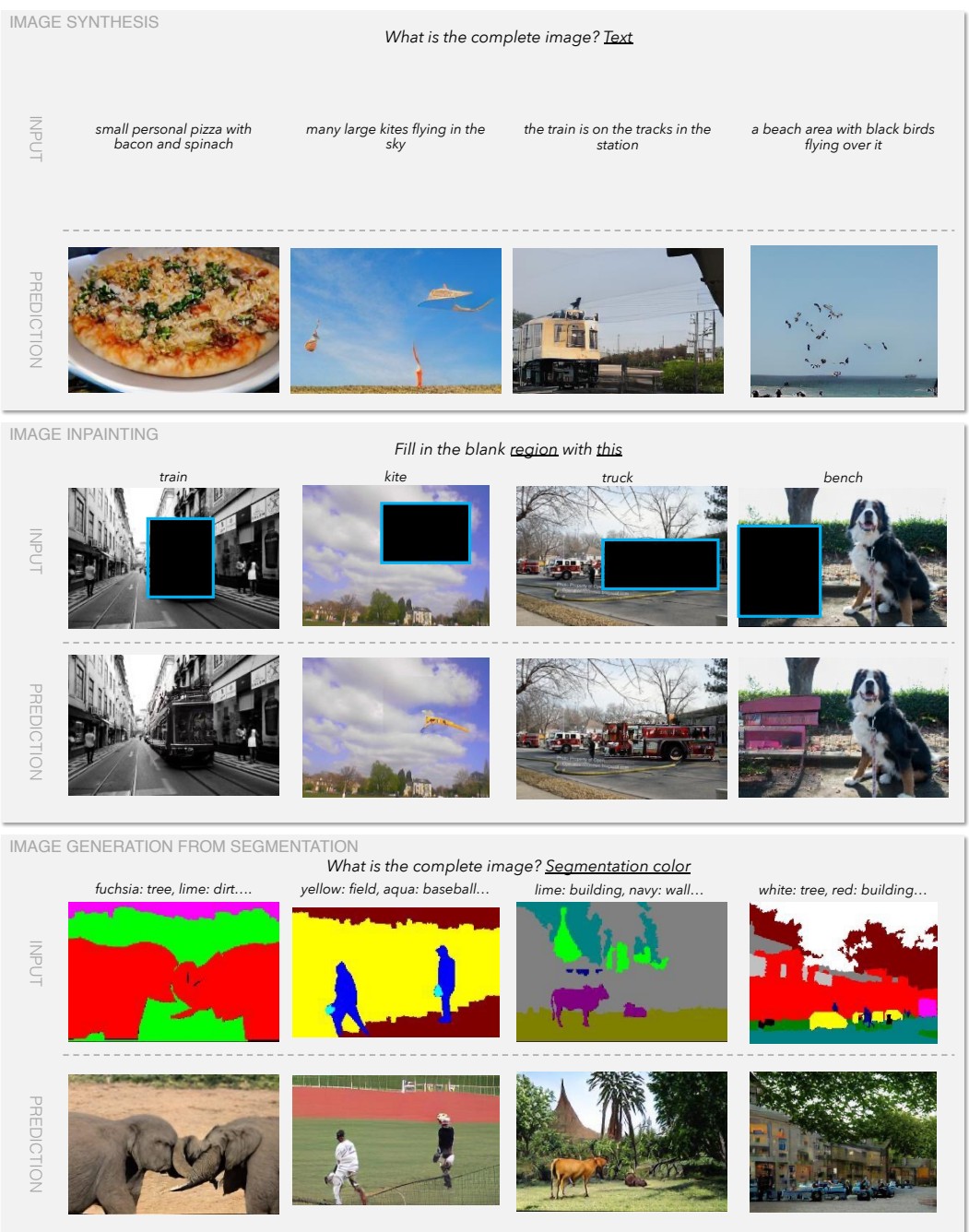

Figure 6: Image synthesis qualitative examples.

et al., 2021; Li et al., 2022b; 2021) or auto-regressive generation losses (Wang et al., 2022d;a; Yu et al., 2022a), have become common. Several works have also directly used object detection or segmentation datasets for pre-training Yuan et al. (2021); Wang et al. (2022b); Sun et al. (2022). The generalized masked-data-modeling objective used in UNIFIED-IO is similar to the ones used in several recent works (Wang et al., 2022c; Peng et al., 2022; Singh et al., 2022).

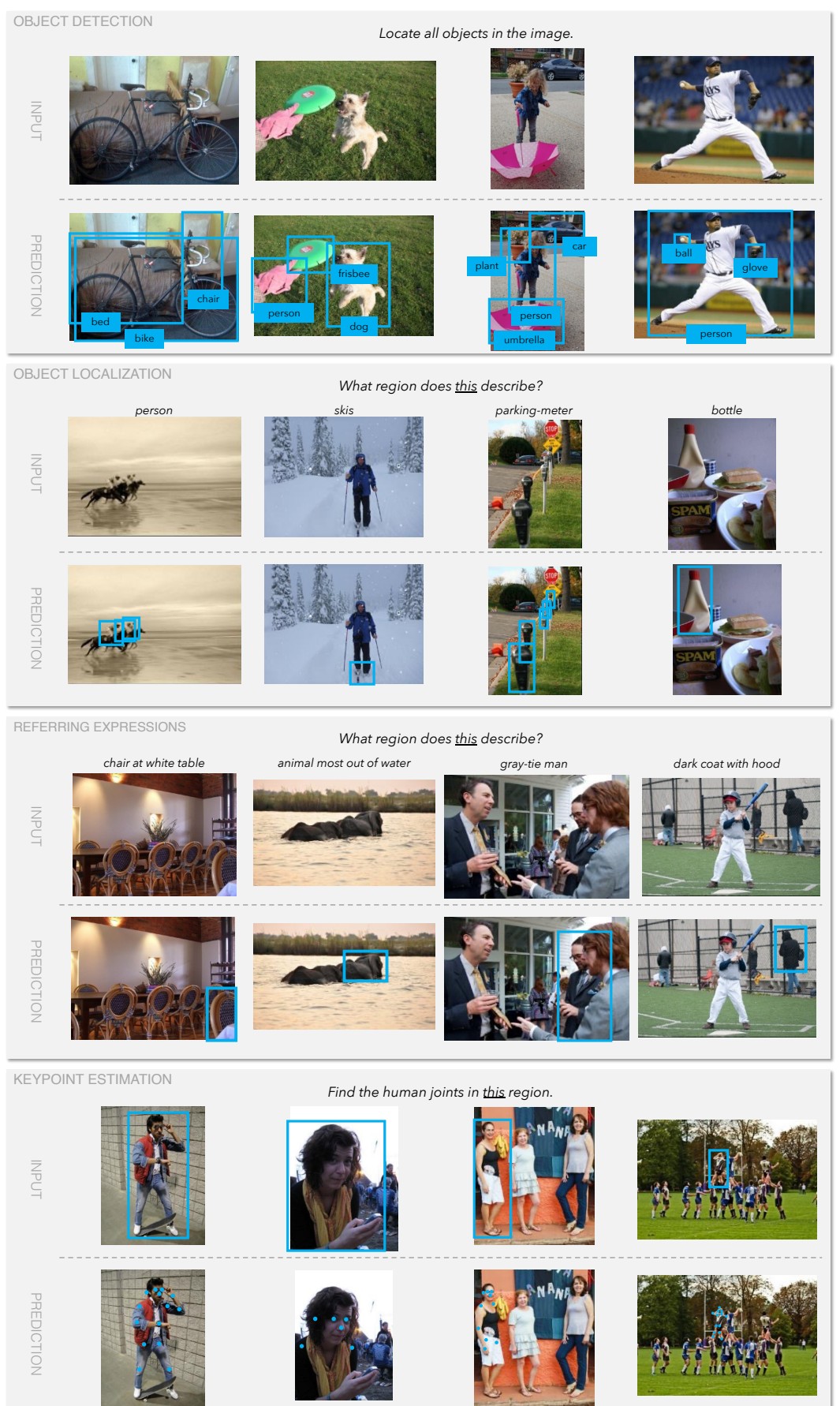

Figure 7: Sparse labelling qualitative examples.

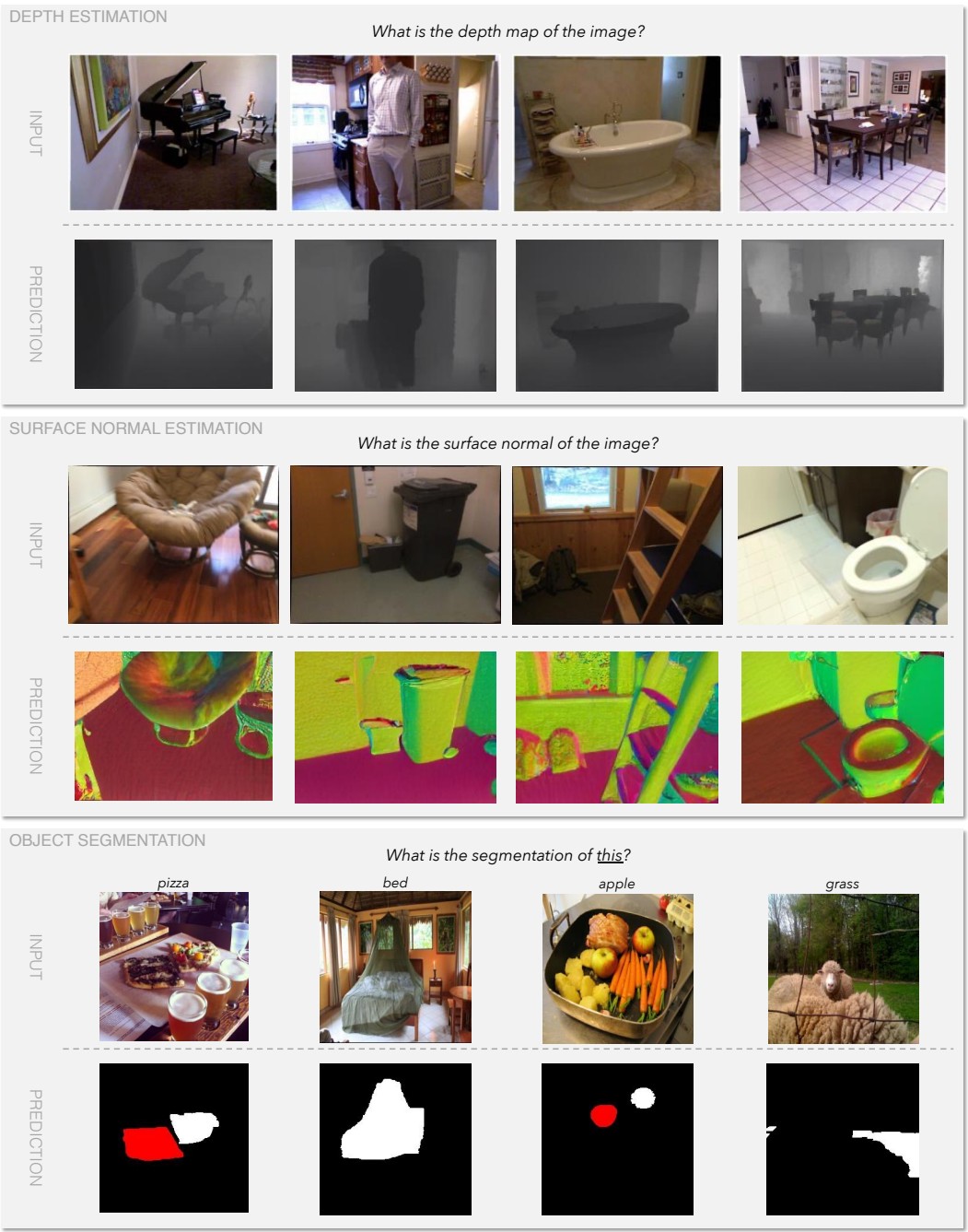

Figure 8: Dense labelling qualitative examples.

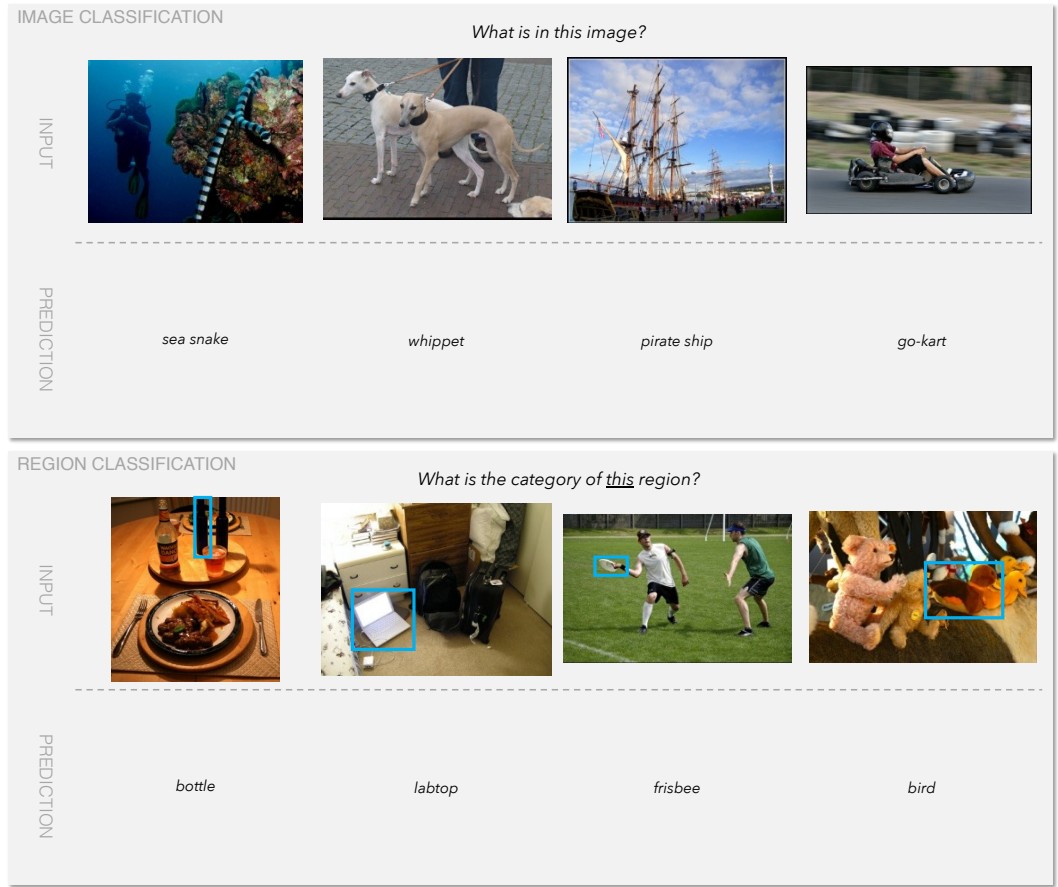

Figure 9: Image classification qualitative examples.

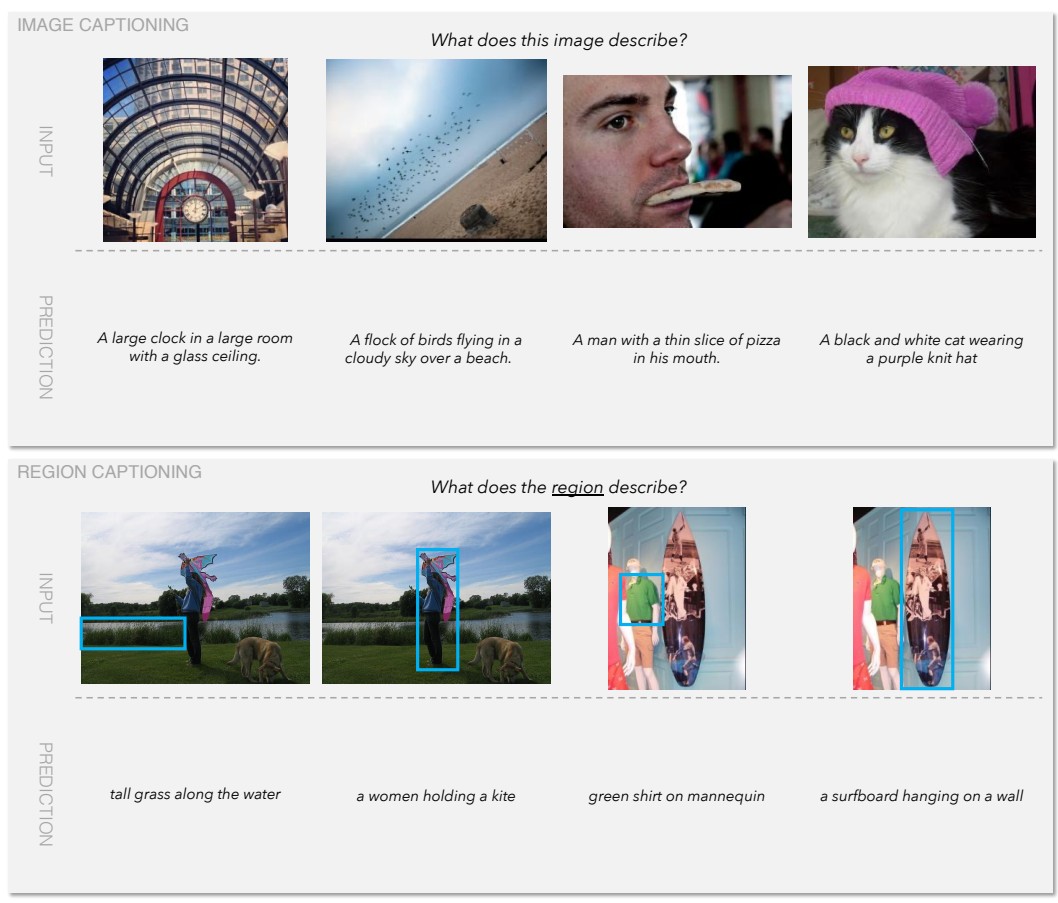

Figure 10: Image captioning qualitative examples.

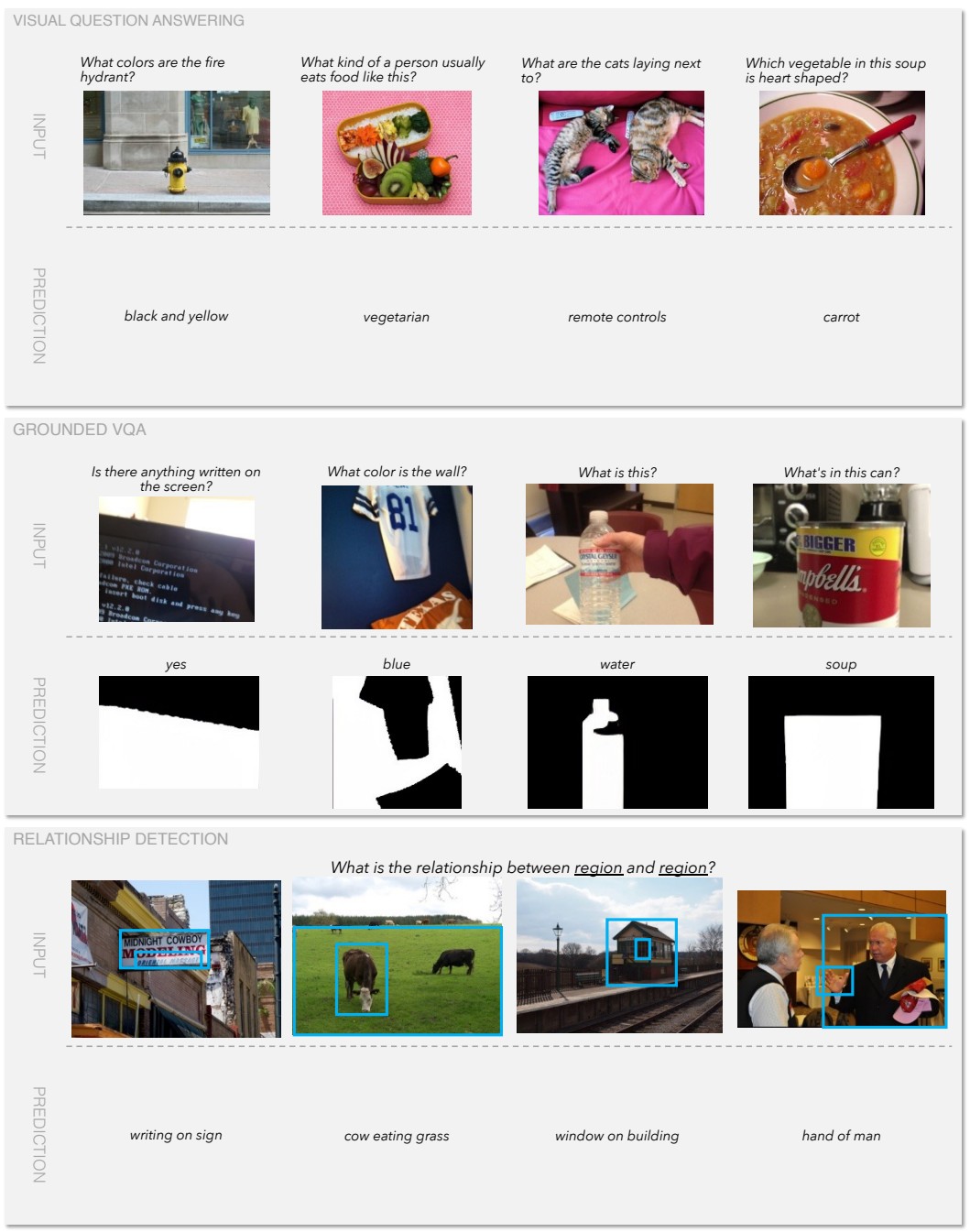

Figure 11: Vision and language qualitative examples.

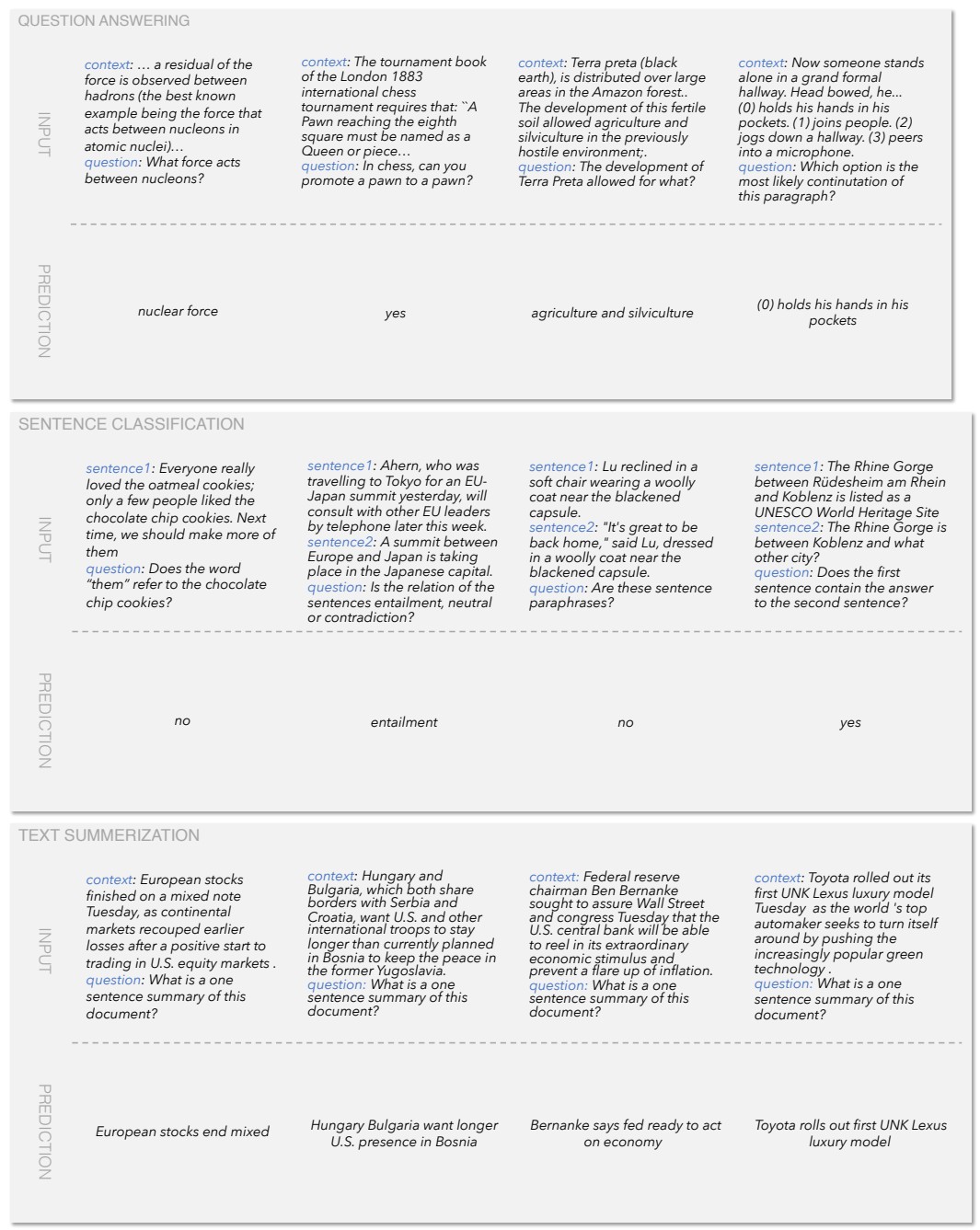

Figure 12: Natural language processing qualitative examples.

