# OpenReview forum: "UNIFIED-IO: A Unified Model for Vision, Language, and Multi-modal Tasks"
_ICLR.cc/2023/Conference — ICLR 2023 notable top 25%_

### Official Review · Reviewer_iS57 · 2022-10-19

**Confidence:** 5
**Correctness:** 4
**Technical Novelty And Significance:** 3
**Empirical Novelty And Significance:** 3
**Recommendation:** 8

**Clarity, Quality, Novelty And Reproducibility:**

The work is clearly presented. The concept is not novel, but the execution of the concept is better than existing methods. The implementation details are provided, but the code and model do not seem to be open-sourced.

**Strength And Weaknesses:**

Strength:
- The number of tasks supported by Unified-IO is impressive. It would require an enormous amount of engineering effort to unify the data format for tasks of diverse nature, not to mention setting up the multi-task training. The community would benefit a lot if the authors could open-source their code for data preprocessing and multi-task training.
- It is interesting to see that a unified model can achieve competitive performance on multiple tasks without per-task finetuning. In particular, Table 4 sheds some light on how various tasks may benefit or conflict each other.
- The paper is very well-written with a clear structure and flow.


Weaknesses:
- In NLP, task unification enables **zero-shot** generalization to novel tasks unseen during training. Papers such as Flamingo have also shown that models trained on image captioning can solve zero-shot VQA. Unified-IO shows the capability to generalize to novel concepts, but concept-generalization is much easier than task-generalization. Have the authors observed any capability of Unified-IO to generalize across tasks? This might be the most important advantage that is expected from task unification.
- Besides reducing multiple models to a single one, what other advantage does multi-task learning bring? Table 4 shows that different tasks may benefit or harm each other. It would be good to see a more principled analysis of task relationships. If one is interested in a particular task, what is the optimal multi-task selection strategy?
- How important is the pre-training stage? It would be good to see some ablation experiments to study the effect of pre-training and analyze each proposed pre-training objective.
- The paper claims the pre-training stage to be ``webly supervised''. However, Imagenet21k has human-annotated class labels.

**Summary Of The Paper:**

This paper proposes Unified-IO, a single Seq2Seq transformer model that can perform a wide range of tasks in vision, NLP, and vision-language domains. The unification of tasks is achieved by converting the input&output of each task into a sequence of discrete tokens. The model is trained in two stages: pre-training with masked language/image denoising and multi-task learning. The resulting model achieves competitive performance on downstream tasks without per-task finetuning.

**Summary Of The Review:**

Unified-IO demonstrates the possibility of unifying a wide range of different tasks into a single model with competitive performance. This contribution alone can justify acceptance in my opinion. The paper is a bit lacking in terms of more in-depth anaylsis of pre-training and multi-task learning. The work will also be much more valuable if the code and model can be open-sourced.

---

> ### Author Response · Authors · 2022-11-19
> **Response to Reviewer iS57**
>
> Thanks for your time reviewing and providing detailed suggestions for our paper! We have updated the paper according to your suggestions and addressed all concerns below.
>
> 1. **Have the authors observed any capability of Unified-IO to generalize across tasks? This might be the most important advantage that is expected from task unification. Besides reducing multiple models to a single one, what other advantage does multi-task learning bring?**
>
> We agree that cross-task generalization is a key direction for future work. As suggested by Reviewer 6qFN, we did some preliminary exploration on animal and human surface normal estimation (see the review response and Section A.7, also see A.6 for a study on the related topic of prompt generalization). Multi-tasking additionally offers the potential to transfer knowledge between tasks in a way similar to humans. While our study in Table 4 was not able conclusive demonstrate this, we still think this is an important goal for the community to work towards and our work expanding the range of tasks that can be done by a single model will help facilitate progress in that direction.
>
> 2. **Table 4 shows that different tasks may benefit or harm each other. It would be good to see a more principled analysis of task relationships. If one is interested in a particular task, what is the optimal multi-task selection strategy?**
>
> We agree with the reviewer that this is a very interesting topic that deserves more investigation. However, having a principled analysis of task relationships on massive vision, language, and V&L tasks requires many ablations and extensive amounts of computing which is beyond our current capacity. We will leave a more principled analysis of task relationships in future work.
>
> 3. **How important is the pre-training stage? It would be good to see some ablation experiments to study the effect of pre-training and analyze each proposed pre-training objective.**
>
> Pre-training stage is very important. We train the XL model on GRIT-related tasks from scratch and pretrained checkpoints. We report the 50k step results and will update the final version of the paper. As can be seen in the table, pretraining significantly improves performance overall (7.54 vs. 43.04).
>
> | Method | Categorization | Localization | VQA | Refexp | Segmentation | Keypoint | Normal | All |
> | --------- | :--: |  :--: | :---------: | :------: | :------: |  :------: | :------: | :------: |
> |Scratch | 0.37 | 17.59 | Failed | 5.20 | Failed | 3.17 | 26.44 | 7.54 |
> |Pretrain | 54.26 | 61.80 | 0.03 | 74.84 | Failed | 71.55 | 38.76 | 43.04 |
>
> 4. **The paper claims the pre-training stage to be ``webly supervised''. However, Imagenet21k has human-annotated class labels.**
>
> Thanks for pointing this out. LIAON-400M dataset is not available during the time of development of our model. So, we ensemble the publicly accessible dataset with a smaller scale. We removed the ``web supervised'' in the pre-training stage.
>
> 5. **The work will also be much more valuable if the code and model can be open-sourced.**
>
> Thanks! As promised in the paper, we will release all the source code and pre-trained models to the public.

---

> > ### Comment · Reviewer_iS57 · 2022-11-20
> > **Thanks for the response**
> >
> > Thanks for the authors' response! I will remain my score of acceptance.

---

### Official Review · Reviewer_6qFN · 2022-10-20

**Confidence:** 4
**Correctness:** 4
**Technical Novelty And Significance:** 2
**Empirical Novelty And Significance:** 3
**Recommendation:** 6

**Clarity, Quality, Novelty And Reproducibility:**

The paper is clearly written, and is the first implementation of a simple framework that can handle a large variety of vision and language tasks.

**Details Of Ethics Concerns:**

I have no major ethics concern for this work. The work is trained on public datasets. If the model is used to generate text and images, then it is subject to the same risk as most language models and image generation models.

**Strength And Weaknesses:**

**Strengths:**
1. The method unifies input and output of V&L tasks to sequences in a clean and relatively simple way.
2. The main architecture that follows T5 is also conceptually simple and clean, consisting of a transformer encoder and decoder without much tweaking/modification.
3. No task-specific augmentations are used.
4. It is non-trivial in terms of engineering to combine so many datasets and tasks together, as well as evaluation on a large variety of tasks.
5. This work is a demonstration of clever combination of existing techniques into a generic framework. The framework only consists of a few well-established components (T5, Seq2Seq, VQ-VAE as used in UViM, masked token prediction, coordinates tokenization as proposed by Pix2Seq) while obtaining decent performance.

**Weaknesses:**
1. The need for VQ-VAE may limit the quality of image synthesis (the model is only trained on image synthesis but not evaluated on such tasks).
2. The model is only evaluated on two dense output tasks: depth estimation and segmentation (in GRIT).
  - For depth estimation, it is compared with UViM but there is no comparison of model size.
  - For segmentation, there is only a mask R-CNN baseline for GRIT (without model size comparison), but there are better-established benchmarks (such as COCO, Cityscape) where more methods have reported on.
  - Results on more dense output tasks such as panoptic segmentation would be interesting.
3. For tasks that require coordinates generation, the model seems to do better in short-length or fixed-length outputs (e.g. keypoint), but not in free-length outputs such as object detection. There is also no evaluation on certain core vision tasks such as object detection, even if the model is trained on it.
4. Overall, the benchmarks could provide more comparison to be more convincing. Right now the results shows that UNIFIED-IO can do a large variety of tasks, but only some of them competitively.
  - GRIT is relatively new, so it would be nice to evaluate on salient benchmarks on some of the tasks without much comparison in GRIT (e.g. segmentation).
  - Additional tasks are mostly with text outputs or image classification, except one task for depth estimation. More tasks with coordinates generation and dense output would be nice.
  - No model comparison in Table 3 and Table 5, so it is hard to separate gains from model scaling and methodology.
5. Known concern for speed of autoregressive generation.
6. Strictly speaking, all components and techniques have been proposed in previous or concurrent work, so it is a bit weak on novelty.

**Actionable items:**
- More baseline on tasks with coordinates output (object detection) or dense output (segmentation) on popular benchmarks such as COCO. I’d suggest that have object detection, semantic and/or panoptic segmentation, keypoint detection on COCO validation/test set for easier comparison with existing methods and concurrent work.
- Add model size comparison in Table 3 and 5 to help explain how much gain comes from model scaling vs methodology. The XL version with 2.9B parameters is likely much larger than most of the methods in comparison.


**Summary Of The Paper:**

This paper proposes UNIFIED-IO, a unified model for a large variety of vision, language and V+L tasks. By formulating all inputs as sequences of embeddings, and all outputs as sequences of discrete tokens, UNIFIED-IO is able to use a simple Seq2Seq model to handle most V&L tasks such as image synthesis, depth estimation, object detection, segmentation, VQA, and more. It uses the T5 architecture as the Seq2Seq model, the SentencePiece tokenizer for encoding/decoding text tokens, a pretrained VQ-GAN for decoding image tokens into images. The model is trained with two stages: firstly pretrained unsupervised with masked token prediction objective on image, text and image-text pair data, then trained with supervision on a large number of tasks and datasets. The model is then evaluated on the recently proposed GRIT benchmark and 16 additional tasks to show competitive results on most of these tasks without finetuning for any specific task.

**Major contributions of this paper:**
- UNIFIED-IO is the first model to combine a large number of V&L tasks.
- It shows the capability of a transformer Seq2Seq model to learn and perform a large variety of V&L tasks.


**Summary Of The Review:**

This paper proposes a simple and generic framework that can perform on a large variety of vision, language and V+L tasks. Even though it is not topping in technical novelty, it is a good proof of concept, and the amount of tasks and datasets combined is impressive. In addition it is not trivial to combine so many datasets and tasks altogether in terms of engineering. I would recommend it to be accepted, with some reservation, mainly in terms of evaluation of the model. I am happy to change my score if the evaluation of the model is more rounded (on a more balanced set of tasks, on salient benchmarks, for easier comparison with existing methods).

---

> ### Author Response · Authors · 2022-11-19
> **Response to Reviewer 6qFN (1/2)**
>
> Thanks for your time reviewing and providing detailed suggestions for our paper! We have updated the paper according to your suggestions and addressed all concerns below.
>
> 1. **The need for VQ-VAE may limit the quality of image synthesis.**
>
> We agree with the reviewer that VQ-VAE may limit the quality of image synthesis. As discussed in Section 4.4, our approach still has limited image generation capabilities. However, recent work [1] showed that the VQ-VAE with encoder-decoder architecture could achieve state-of-the-art performance on image generation (compared to DALLE2 and the latest diffusion models) but was not available at the time of development.
>
> [1] Jiahui Yu, Yuanzhong Xu, Jing Yu Koh, Thang Luong, Gunjan Baid, Zirui Wang, Vijay Vasudevan, Alexander Ku, Yinfei Yang, Burcu Karagol Ayan, et al. Scaling autoregressive models for a content-rich text-to-image generation. arXiv preprint arXiv:2206.10789, 2022b
>
> 2. **For depth estimation, it is compared with UViM, but there is no comparison of model size.**
>
> UViM doesn’t explicitly list the number of parameters in their model. From the paper, they use ViT-L/16 for the encoder and ViT-B for the decoder with 4096 dictionary entries, which contain approximately 400M parameters – which lie in the middle of our base model (241M) and Large model (776). Different from UViM, we directly use the VQ-GAN model checkpoints trained on the imagenet. Thus there is no depth image involved when training the VQ-GAN. More importantly, our base model is a multi-task model that can handle massive vision, language, and vision & language tasks. Due to the large variance in dataset size, some of the tasks are rarely sampled – the depth estimation task sample rate is only 0.43% during training, as illustrated in Appendix A.3. In contrast, UViM is specifically trained for depth estimation.
>
> 3. **No model comparison in Table 3 and Table 5, so it is hard to separate gains from model scaling and methodology. Add model size comparison in Table 3 and 5 to help explain how much gain comes from model scaling vs methodology. The XL version with 2.9B parameters is likely much larger than most of the methods in comparison.**
>
> Due to space limitations, we listed the number of parameters for each model in GRIT benchmark in supplementary materials. (Table 6 of A.5 in Appendix). We recommend the reviewer check A.5 for details.
>
> Upon the reviewer’s suggestion, we list all the methods with parameter size for table 5. If there is no explicit list of the number of parameters in the original paper, we estimate the number of parameters based on their model descriptions and use ~  in the table.
>
> | Method |  UViM | BinsFormer | COCA | MAE | KAT | GPV2 | Flamingo | JSL | OFA | SVT | T5 | PaLM | Turing NLR | ST-MOE | DeBERTa |
> | ---------- | :------: | :---------: | :---: | :-------: | :------: | :----: | :----: | :---: | :---: | :----: | :----: | :----: |  :----: |  :----: |  :----: |
> | #Params | ~400M | ~200M | 2.1B |  ~300M | ~175B* | 370M | 80B | 108M |  473M | ~1.5B | 11B | 540B | 5.4B | 32B | 1.5B |
>
> *: KAT requires access to GPT-3 model, so we list ~175B in the table.
>
> From the table, we can see a lot of the method in comparison is larger than our XL model. There are other factors that need to be considered as well, such as data augmentation, input image size, different training data, etc.
>
> 4. **GRIT is relatively new, so it would be nice to evaluate on salient benchmarks on some of the tasks without much comparison in GRIT (e.g. segmentation).**
>
> While GRIT was introduced in the last 12 months, it isn't new data per se. It leverages well-known and well-studied data sources. However, since it does evaluate the performance, robustness, and calibration of a vision system across a variety of image prediction tasks, concepts, and data sources for a single model, it is perfect for our goal – which is to build a single unified model that can support a diverse set of tasks across computer vision and language with little to no need for task-specific customizations and parameters rather than achieve state-of-the-art performance on each task.
>
> GRIT uses sources from existing datasets such as COCO, VQA v2, and NYU v2. As shown in Table 2 of the GRIT paper, the same concept of localization, segmentation, and keypoints tasks are only constructed from COCO datasets. Thus it is fair to compare it with the well-known baseline Mask RCNN. Due to space constraints, table 6 in the appendix shows the results of the same concept and the new concept. Our approach achieves competitive performance compared to Mask RCNN. Since our paper is one of the early attempts to develop one single unified model for major vision, vision, and language & language tasks, we left the improvements over the individual tasks as future work.

---

> > ### Author Response · Authors · 2022-11-19
> > **Response to Reviewer 6qFN (2/2)**
> >
> > 5. **For tasks that require coordinates generation, the model seems to do better in short-length or fixed-length outputs (e.g. keypoint), but not in free-length outputs such as object detection. There is also no evaluation on certain core vision tasks such as object detection, even if the model is trained on it.**
> >
> > We mainly evaluate coordinate generation tasks on GRIT. As discussed in the previous question, GRIT benchmarks with the same concept are only constructed from COCO datasets. So the human keypoint estimation can be counted as core vision tasks in coordinates generation tasks. We recommend the reviewer check A.5 same concepts for more details.
> >
> > As discussed in the limitation section of the paper, for object detection, while Unified-IO generally produces accurate outputs, we find the recall is often poor in cluttered images. Prior work (Chen et al., 2022b) has shown this can be overcome with extensive data augmentation techniques, but these methods are not currently integrated into Unified-IO.
> >
> > 6. **Additional tasks are mostly with text outputs or image classification, except one task for depth estimation. More tasks with coordinates generation and dense output would be nice.**
> >
> > The VizWiz Grounding tasks (VizWizG) also require dense outputs. Please check the Grounded QA of Figure 10 in Appendix for more qualitative results. Figure 5 in Appendix also shows the qualitative results of image generation, image inpainting, and image generation from segmentation. We leave more tasks with coordinates generation and dense outputs for future work.
> >
> > 7. **Known concern for speed of autoregressive generation.**
> >
> > We agree with the reviewer that there are known concerns about the speed of auto-regressive generation, especially for object detection and segmentation. However, the main contribution of this paper is to explore whether a single model without a task-specific head can handle massive, diverse vision, vision & language, and language tasks. This work does not attempt to address the speed issue, which is shared with all existing and shared with other existing unified models (such as UViM etc.).
> >
> > 8. **Strictly speaking, all components and techniques have been proposed in previous or concurrent work, so it is a bit weak on novelty.**
> >
> > We respectfully disagree with the reviewer’s opinion. The components and techniques proposed in concurrent work should not undermine our work’s novelty. We are one of the first few papers that propose to use an auto-regressive model for dense labeling tasks such as segmentation, depth estimation, and surface normal estimation. More importantly, we showed a single model without a task-specific head can handle massive, diverse vision, V&L language, and language tasks. We kindly request the reviewer exclude the concurrent work when evaluating the novelty of our paper.

---

> > > ### Comment · Reviewer_6qFN · 2022-11-22
> > > **Thanks for the response.**
> > >
> > > Thanks for the authors' detailed response. I agree that concurrent work should not have been considered in the review. However I do maintain my opinion that the components and techniques have already been proposed before, like another reviewer said, the concept is not novel, but the execution of the concept is better than existing methods. I do appreciate the execution, like I mentioned in my original review (strength #4), and for that I do support this paper be accepted.

---

### Official Review · Reviewer_gX6J · 2022-10-25

**Confidence:** 4
**Clarity, Quality, Novelty And Reproducibility:** The overall clarity, quality, and nov…
**Correctness:** 4
**Technical Novelty And Significance:** 3
**Empirical Novelty And Significance:** 3
**Recommendation:** 8

**Strength And Weaknesses:**

Strength:
The idea of building a unified model for vision and language based on a powerful sequence-to-sequence model has been around recently, but the paper greatly scaled it up to a large number of tasks and datasets, nicely demonstrating that it can achieve decent performance on general vision tasks without task-specific inductive biases. The implication of this result is huge in my opinion.

Weakness:
I do not have a major concern in general since the experiments and results are strong and convincing. One weakness observed from Table 4 is that the multi-task pre-training often hinders the performance of individual downstream tasks, which implies that improving the performance of the model is not simply about scaling up the pre-training data but requires more careful considerations in choosing the pre-training tasks and data. However, I believe that this is not the main scope of this work hence do not have a major concern. Other than that, I would like to suggest authors include some discussions and results that might help readers to understand the limitation/capability of this work better, which are summarized below.

1. Impact of multi-task learning

Other than efficiency, one of the benefits of having a single parameterization of multiple tasks is that it might learn to understand tasks across domains. For instance, some tasks used in the pre-training have strong biases in their domains (e.g., pose is mostly labeled for humans and the surface normal is labeled mostly for rigid indoor objects). Yet, an ideal multi-task learner might be able to disentangle the task and domain, and perhaps generalize the task across domains in other tasks. Many works on the vision-language model often demonstrate this out-of-distribution generalization in a limited context (e.g., in the context of open-vocabulary learning where the pre-trained model is applied to downstream tasks of unseen object categories), but it would be interesting to see how much a massive multi-task learner such as the proposed model can extend it to more difficult cases (e.g., estimating a pose of animal or surface normal of human).


2. Performance gap in zero-shot generalization

The authors demonstrated that the proposed method exhibits compelling performance to the SOTA models specialized in individual tasks. Although it is quite encouraging, it does not directly show the upper-bound performance of the proposed model on individual tasks (i.e., how powerful the architecture and pre-training strategy for each task and how much performance we need to compromise by eliminating the test-time finetuning). Ablation studies with and without fine-tuning in downstream tasks might be helpful in understanding this gap.


**Summary Of The Paper:**

This paper proposed a unified learner for various vision and language tasks that involve dense and sparse prediction. To be able to represent languages and various vision tasks in a homogeneous way, the authors employed the tokenized representation where the discrete codebooks for words and visual patches are learned by the VQ-GAN framework and used in the sequence-to-sequence model. When learned with a gigantic dataset of visual and linguistic tasks, the proposed model demonstrated outstanding performance on various downstream tasks without task-specific retraining.

**Summary Of The Review:**

In general, I am positive about this paper due to the strong and convincing evaluation results. Although the technical contribution of this work is not ground-breaking, the authors executed the idea of building a unified model for vision and language based on a powerful sequence-to-sequence model well and achieved encouraging results.

---

> ### Author Response · Authors · 2022-11-19
> **Response to Reviewer gX6J**
>
> Thanks for your time reviewing and providing detailed suggestions for our paper! We have updated the paper according to your suggestions and addressed all concerns below.
>
> 1. **Impact of multi-task learning. it would be interesting to see how much a massive multi-task learner such as the proposed model can extend it to more difficult cases (e.g., estimating a pose of animal or surface normal of human).**
>
> Thanks, we agree with the reviewer that the benefit of having a single parameterization of multiple tasks is it might learn to understand tasks across domains. In section 4.2, we conduct a preliminary study of the impact of multi-task learning by leaving out individual task groups from multi-task training. For more complex cases, we follow the reviewer’s suggestion and use XL model to estimate the pose of animals or surface normal of human and other types of scenes.
>
> Appendix A.7 and Figure 5 show the qualitative examples of these tasks. For surface normal detection, we find that the model produces plausible images even for objects like cats or humans that are not in the training data. However, other scenes, such as outdoor scenes (Figure 5 bottom right), are less coherent. For animal pose estimation, our model is able to find animal keypoints at least partially. For animals standing or crouching on two legs, the keypoints are reasonably accurate (first two images), however for animals standing on four legs, the model will find leg and eye points but then guess arm positions that would make sense for a person instead of attaching points to the other legs.
>
> Since this work is one of the early attempts to develop one single unified model for almost all major vision tasks and some NLP tasks, we leave a more comprehensive study of the impact of multi-task learning as future work.
>
> 2. **Performance gap in zero-shot generalization. Ablation studies with and without fine-tuning in downstream tasks might be helpful in understanding this gap.**
>
> The performance of our model is largely affected by the training recipes (e.g., data augmentation, initialization, optimization, etc.) of different algorithms. Unlike most of the concurrent work, our model largely follows the design of T5, and we only make a few architectural changes to adapt to our settings. Unified-IO is still at an early stage of this line of research. We only focus on building a single unified model in this paper. Due to time and computing constraints, we finetune the large and XL model on the imagenet 2012 dataset to help understand the gap with or without finetuning.
>
> We finetune the large and xl model on the imagenet 2012 dataset with batch size 256 and constant learning rate of 0.0003 for 50000 steps. Similar to the multi-tasking setup, we do not use extensive data augmentations such as randAug and CutMix, no dropout in the model. We use a beam size of 4 during inference and do not constrain the vocabulary during training and evaluation.
>
> | Method |  Multi-Tasking | Finetuning | Gain|
> | --------- | :--: | :---------: | :------: |
> |Unified-IO$_{LARGE}$ | 71.8 | 78.0 | +6.2 |
> |Unified-IO$_{XL}$ | 79.1 | 83.2 | +4.1 |
>
> The above table shows performance gain after the finetuning. We can see the large model has relatively more gain compared to the XL model (+6.2 vs. +4.1), and the XL model reaches 83.2 after finetuning. Although still behind SOTA, our model can benefit from finetuning and further improved with task-specific training recipes. Finding a balanced training recipe that works well on multiple modalities without sacrificing the performance of an individual task will leave for future work.

---

> > ### Comment · Reviewer_gX6J · 2022-11-23
> > **Thanks for the responses**
> >
> > I appreciate the authors for their thorough response, especially about the cross-domain generalization case study. The additional results are really helpful in understanding the capability and limitations of the proposed method. I believe that this work is worth reading by the community and recommend acceptance.

---

### Official Review · Reviewer_csLY · 2022-10-25

**Confidence:** 4
**Correctness:** 4
**Technical Novelty And Significance:** 3
**Empirical Novelty And Significance:** 4
**Recommendation:** 8

**Clarity, Quality, Novelty And Reproducibility:**

- **Clarity**: The paper is well-organized and overall easy to follow.

- **Quality**: Thorough experiments are conducted to validate the proposed framework. The presentation is clear. Most related methods have been discussed and compared.

 - **Novelty**: The technical novelty of the proposed method is relatively low. The whole framework is new.

- **Reproducibility**: It would be difficult to fully reproduce the results considering the missing details and requirements of enormous computational resources. It would be appreciated if the authors could make the code/pre-trained models public which might be very useful for future research and applications.

**Strength And Weaknesses:**

Strengths:

- The paper presents one of the early attempts to develop one single unified model for major vision tasks including classification, dense predictions, locations, and vision-language tasks and some NLP tasks. The method also achieves top performance on GRIT benchmark and comparable results with specialized state-of-the-art methods.

- The paper provides a practical solution to combine multiple datasets and supervisory signals with discrete encoding and decoding.

- The method shows good scaling ability for very large models.

Weaknesses:

- The technical novelty of the proposed method is somehow limited although the whole framework is new and the results are significant.  The idea of encoding labels to discrete tokens is similar to earlier work UViM. [r1] also uses the VQ-GAN encoder to convert dense labels to discrete tokens.

[r1] Visual Prompting via Image Inpainting, NeurIPS 2022.

- Some implementation details are missing. For example, the training details of the VQ model are not mentioned in both the main paper and the supplementary materials.

Minor issues:

- In Sec. 1 and Fig. 1, it is mentioned a "VQ-VAE" model is used for image serialization. But in Sec. 3, only "VQ-GAN" is mentioned. I understand it seems the VQ-GAN architecture is used but the adversarial loss is not added during encoding images. But different terms in different sections may make the readers hard to follow.

**Summary Of The Paper:**

This paper presents a unified framework for vision, language and multi-modal tasks. Instead of directly predicting labels with different formats, a unified task representation is designed to enable joint training of multiple tasks. The method exhibits state-of-the-art performance on the GRIT benchmarks and comparable results with specialized state-of-the-art methods.

**Summary Of The Review:**

The paper presents one of the early attempts to develop one single unified model for almost all major vision tasks and some NLP tasks. The method clearly outperforms previous frameworks on the GRIT benchmark. Although there is still a notable performance margin between unified models and specialized models, this paper presents a solid attempt toward unified visual perception. Therefore, I recommend accepting this paper.

---

> ### Author Response · Authors · 2022-11-19
> **Response to Reviewer csLY**
>
> Thanks for your time reviewing and providing detailed suggestions for our paper! We have updated the paper according to your suggestions and addressed all concerns below.
>
> 1. **The technical novelty of the proposed method is somehow limited, although the whole framework is new, and the results are significant. The idea of encoding labels to discrete tokens is similar to earlier work UViM. [r1] also uses the VQ-GAN encoder to convert dense labels to discrete tokens.**
>
> Thanks for pointing out the related work [r1]. We add [r1] to the related work section and discuss the difference as compared to our approach. As discussed in the related work section, UViM and Visual Prompting [r1] is concurrent to our work and should not impair our technique novelty. UViM learns the guiding code on the target domain (depth, segmentation, etc.) while we use the VQ-GAN model trained on imagenet. Visual Prompting is trained and evaluated on a new dataset that contains unlabeled figures from academic papers, and there is no evaluation on a standard benchmark such as nyu v2 datasets. More importantly, our approach covers a wide span of tasks and modalities and focuses on multi-tasking rather than task-specific fine-tuning.
>
> 2. **Some implementation details are missing. For example, the training details of the VQ model are not mentioned in both the main paper and the supplementary materials.**
>
> As mentioned in 3.1 (main paper), we directly use the imagenet-pretrained VQ-GAN model in our approach. We will emphasize this detail in the `Implementation Details` section. As promised in the paper, we will release all the source code and pre-trained models to the public.
>
> 3. **VQ-VAE and VQ-GAN in different sections may make the readers hard to follow.**
>
> Thanks for the suggestion. We change all VQ-VAE in the paper to VQ-GAN to make it the reader easier to follow.

---

> > ### Comment · Reviewer_csLY · 2022-11-22
> > **Thanks for the feedback**
> >
> > Thanks for the response. After reading the response and other reviews, I would like to keep my initial rating.

---

### Decision · Program_Chairs · 2023-01-20

**Decision:**

Accept: notable-top-25%

**Justification For Why Not Higher Score:**

Seems interesting but not really ground breaking. But reviewers liked the paper.

**Justification For Why Not Lower Score:**

Seems interesting and pretty good but maybe it will get more interest than a regular poster i guess.

**Metareview: Summary, Strengths And Weaknesses:**

All reviewers liked the paper. Scores are good! Accept! Congrats!!

**Note From Pc:**

if the above contains the word "oral" or "spotlight" please see: "oral" presentation means -> notable-top-5% and "spotlight" means -> notable-top-25%. As stated in our emails, we are disassociating presentation type from AC recommendations